# Understanding Pathologies of Deep Heteroskedastic Regression

Eliot Wong-Toi[1]        Alex Boyd[1]        Vincent Fortuin[2]        Stephan Mandt[1,3]

[1]Department of Statistics, University of California, Irvine, USA
[2]Helmholtz AI, Munich, Germany
[3]Department of Computer Science, University of California, Irvine, USA

## Abstract

Deep, overparameterized regression models are notorious for their tendency to overfit. This problem is exacerbated in heteroskedastic models, which predict both mean and residual noise for each data point. At one extreme, these models fit all training data perfectly, eliminating residual noise entirely; at the other, they overfit the residual noise while predicting a constant, uninformative mean. We observe a lack of middle ground, suggesting a phase transition dependent on model regularization strength. Empirical verification supports this conjecture by fitting numerous models with varying mean and variance regularization. To explain the transition, we develop a theoretical framework based on a statistical field theory, yielding qualitative agreement with experiments. As a practical consequence, our analysis simplifies hyperparameter tuning from a two-dimensional to a one-dimensional search, substantially reducing the computational burden. Experiments on diverse datasets, including UCI datasets and the large-scale ClimSim climate dataset, demonstrate significantly improved performance in various calibration tasks.

## 1 INTRODUCTION

Homoskedastic regression models assume constant (e.g., Gaussian) output noise and amount to learning a function $f(x)$ that tries to predict the most likely target $y$ for input $x$. In contrast, *heteroskedastic* models assume that the output noise may depend on the input features $x$ as well, and try to learn a conditional distribution $p(y|x)$ with non-uniform variance. The promise of this approach is to assign different importances to training data and to train models that "know where they fail" [Skafte et al., 2019, Fortuin et al., 2022].

Unfortunately, overparameterized heteroskedastic regres-

sion models (e.g., based on deep neural networks) are prone to extreme forms of overfitting [Lakshminarayanan et al., 2017, Nix and Weigend, 1994]. On the one hand, the mean model is flexible enough to fit every training datum's target perfectly, while the standard deviation network learns to maximize the likelihood by shrinking the predicted standard deviations to zero. On the other hand, just the tiniest amount of regularization on the mean network will make the model prefer a constant solution. Such a flat prediction results from the standard deviation network's ability to explain all residuals as random noise, thus overfitting the data's empirical prediction residuals. Fig. 1 shows both types of overfitting.

While several practical solutions to learning overparameterized heteroskedastic regression models have been proposed [Skafte et al., 2019, Stirn and Knowles, 2020, Seitzer et al., 2022, Stirn et al., 2023, Immer et al., 2023], no comprehensive theoretical study of the failure of these methods has been offered so far. We conjecture this is because overparameterized models have attracted the most attention only in the past few years, while most classical statistics have focused on under-parameterized (e.g., linear) regression models where such problems cannot occur [Huber, 1967, Astivia and Zumbo, 2019].

This paper provides a theoretical analysis of the failure of heteroskedastic regression models in the overparameterized limit. To this end, it borrows a tool that abstracts away from any details of the involved neural network architectures: classical field theory from statistical mechanics [Landau and Lifshitz, 2013, Altland and Simons, 2010]. Via our field-theoretical description, we can recover the optimized heteroskedastic regressors as solutions to partial differential equations that can be derived from a variational principle. These equations can in turn be solved numerically by optimizing the field theory's free energy functional.

Our analysis results in a two-dimensional *phase diagram*, representing the coarse-grained behavior of heteroskedastic noise models for every parameter configuration. Each of the two dimensions corresponds to a different level of

regularization of either the mean or standard deviation network. As encountered in many complex physical systems, the field theory unveils *phase transitions*, i.e., sudden and discontinuous changes in certain *observables* (metrics of interest) that characterize the model, such as the smoothness of its mean prediction network, upon small changes in the regularization strengths. In particular, we find a sharp phase boundary between the two types of behavior outlined in the first paragraph, at weak regularization.

Our contributions are as follows:

• We provide a unified theoretical description of overparameterized heteroskedastic regression models, which generalizes across different models and architectures, drawing on tools from statistical mechanics and variational calculus.

• In this framework, we derive a field theory (FT), which can explain the observed types of overfitting in these models and describe *phase transitions* between them. We show qualitative agreement of our FT with experiments, both on simulated and real-world regression tasks.

• As a practical consequence of our analysis, we dramatically reduce the search space over hyperparameters by eliminating one parameter. This reduces the number of hyperparameters from two to one, empirically resulting in well-calibrated models. We demonstrate the benefits of our approach on a large-scale climate modeling example.

## 2 PITFALLS OF OVERPARAMETERIZED HETEROSKEDASTIC REGRESSION

**Heteroskedastic Regression** Consider the setting in which we have a collection of independent data points $\mathcal{D} := \{(x_i, y_i)\}_{i=1}^N$ with covariates $x_i \in \mathcal{X} \subset \mathbb{R}^d$ drawn from some distribution $x_i \sim p(x)$ and response values $y_i \in \mathcal{Y} \equiv \mathbb{R}$ normally distributed with unique mean $\mu_i$ and precision (inverse-variance) $\Lambda_i > 0$ (i.e., $y_i \sim \mathcal{N}(\mu_i, \Lambda_i)$). We assume to be in a *heteroskedastic* setting, in which $\Lambda_i$ need not equal $\Lambda_j$ for $i \neq j$. Finally, we assume *both* the mean and standard deviation of $y_i$ to be explainable via $x_i$:

$$y_i \,|\, x_i \sim \mathcal{N}(\mu(x_i), \Lambda(x_i)) \text{ for } i = 1, \ldots, N \quad (1)$$

with continuous functions $\mu : \mathcal{X} \to \mathbb{R}$ and $\Lambda : \mathcal{X} \to \mathbb{R}_{>0}$. In a modeling setting, learning $\Lambda$ can be seen as directly estimating and quantifying the *aleatoric* (data) uncertainty.

**Overparameterized Neural Networks** There exist many options for modeling $\mu$ and $\Lambda$. Of particular interest to many is representing each of these functions as neural networks [Nix and Weigend, 1994]—specifically ones that are overparameterized. These models are well-known *universal function approximators*, which makes them great choices for estimating the true functions $\mu$ and $\Lambda$ [Hornik, 1991].

Let the mean network $\hat{\mu}_\theta : \mathcal{X} \to \mathbb{R}$ and precision network $\hat{\Lambda}_\phi : \mathcal{X} \to \mathbb{R}_{>0}$ be arbitrary depth, overparameterized

feed-forward neural networks parameterized by $\theta$ and $\phi$ respectively. For a given input $x_i$, these networks collectively represent a corresponding predictive distribution for $y_i$:

$$\hat{p}(y_i \,|\, x_i) := \mathcal{N}(y_i; \hat{\mu}_\theta(x_i), \hat{\Lambda}_\phi(x_i)). \quad (2)$$

**Pitfalls of MLE** Our assumed form of data naturally suggests training $\hat{\mu}_\theta$ and $\hat{\Lambda}_\phi$, or rather learning $\theta$ and $\phi$, by minimizing the cross-entropy between the joint data distribution $p := p(x, y) = p(y \,|\, x)p(x)$ and the induced predictive distribution $\hat{p} := \hat{p}(y \,|\, x)p(x)$. This objective is defined as

$$\mathcal{L}(\theta, \phi) := H(p, \hat{p}) = -\mathbb{E}_p \left[ \log \hat{p}(x, y) \right] \quad (3)$$
$$= \int_\mathcal{X} p(x) \int_\mathcal{Y} p(y \,|\, x) \log \mathcal{N}(y; \hat{\mu}_\theta(x), \hat{\Lambda}_\phi(x)) dy dx + c,$$

where $c$ is a constant with respect to $\theta$ and $\phi$. This expectation is often approximated using a Monte Carlo (MC) estimate with $N$ samples, yielding the following tractable objective function:

$$\mathcal{L}(\theta, \phi) \approx \frac{1}{2N} \sum_{i=1}^N \hat{\Lambda}_\phi(x_i)\hat{r}(x_i)^2 - \log \hat{\Lambda}_\phi(x_i), \quad (4)$$

where $\hat{r}(x_i) = \hat{\mu}_\theta(x_i) - y_i$. Minimizing this cross-entropy objective function with respect to parameters $\theta$ and $\phi$ using data samples is synonymous with maximum likelihood estimation (MLE).

Unfortunately, given an infinitely flexible model, this objective function is ill-posed. Our first observation is that, for any non-zero $\hat{\Lambda}_\phi$, we can find a solution for the parameters $\phi$ in the absence of any regularization since the first term in Eq. (4) is minimized when $\hat{\Lambda}_\phi \to 0$, while the second term is minimized when $\hat{\Lambda}_\phi \to \infty$. However, the interplay between $\phi$ and $\theta$ leads to divergences in the absence of any regularization on $\theta$. Without such regularization, the mean function $\hat{\mu}_\theta$ will estimate $y$ perfectly (or rather to arbitrary precision) for at least a single data point $(x_i, y_i)$. Once this happens, the residual for this input $\hat{\mu}_\theta(x_i) - y_i$ approaches zero, and the implicit regularization for $\hat{\Lambda}_\phi$ vanishes, leading $\hat{\Lambda}_\phi(x_i)$ to diverge to infinity. Intuitively, the model becomes infinitely (over-)confident in its prediction for this data point. Once training has reached this point, the objective function becomes completely unstable due to effectively containing a term whose limit naïvely yields $\infty - \infty$.[1]

---

[1] Note that this is predicated on the model being flexible enough to allow for large changes in predictions $\hat{\mu}_\theta(x)$ and $\hat{\Lambda}_\phi(x)$ after iteratively updating parameters $\theta$ and $\phi$ while allowing for minimal changes in neighboring predictions (i.e., $\hat{\mu}_\theta(x')$ and $\hat{\Lambda}_\phi(x')$ for some $x' \in \mathcal{X}$ such that $0 < ||x - x'|| < \epsilon$).

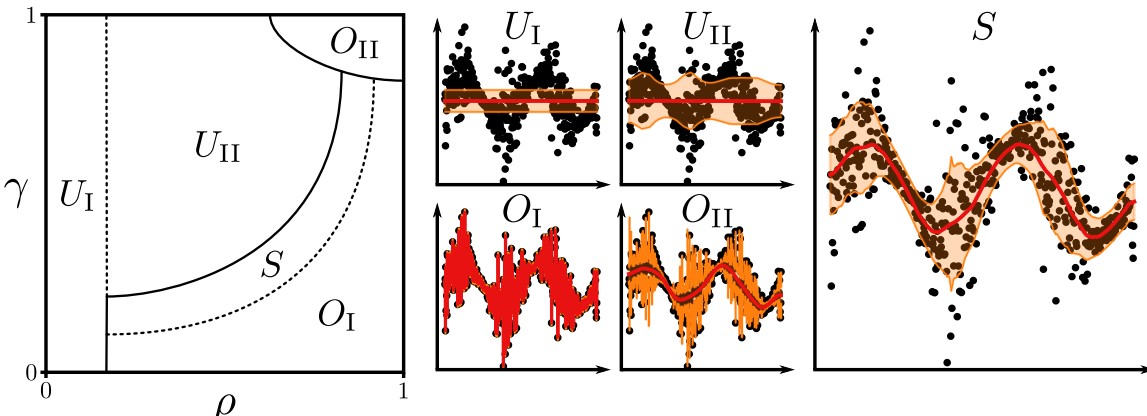

Figure 1: Visualization of a typical phase diagram in $\rho - \gamma$ regularization space for a heteroskedastic regression model (left). Solid and dotted lines indicate sharp and smooth transitions in model behavior respectively. Example model mean fits shown in red (with pointwise $\pm$ standard deviation in orange) from the FT for each key phase (middle and right).

**Regularization** Even though $\hat{\Lambda}_\phi$ is implicitly regularized in the standard cross-entropy loss as mentioned earlier, we posit that additional regularization on $\hat{\Lambda}_\phi$, or rather $\phi$, is required to avoid this instability. It can be tempting to think that one must regularize $\theta$ in order to avoid overfitting. And while this is generally true, the objective function $\mathcal{L}$ will still be unstable so long as *at least* one input $x_i$ yields a perfect prediction (*i.e.*, $y_i = \hat{\mu}_\theta(x_i)$). This situation is still fairly likely to occur even in the most regularized mean predictors and cannot be avoided, especially if $\{y_i\}$ is zero-centered.

To prevent this from happening, we can include $L_2$ penalty terms for both $\theta$ and $\phi$ in our loss function:

$$\mathcal{L}_{\alpha,\beta}(\theta, \phi) := \mathcal{L}(\theta, \phi) + \alpha||\theta||_2^2 + \beta||\phi||_2^2, \quad (5)$$

where $\alpha, \beta > 0$ are penalty coefficients. Intuitively, the primary purpose of regularizing $\theta$ is to prevent the mean predictions from overfitting while the goal of regularizing $\phi$ is to provide stability and control complexity in the predicted aleatoric uncertainty. As $\alpha \to \infty$, the network models a constant mean and, symmetrically, as $\beta \to \infty$ the network models a constant standard deviation. That is, we effectively arrive at a homoskedastic regime as $\beta \to \infty$.[2]

**Reparameterized Regularization** We introduce an alternative parameterization of the regularization coefficients:

$$\mathcal{L}_{\rho,\gamma}(\theta, \phi) := \rho\mathcal{L}(\theta, \phi) + \bar{\rho}\left[\gamma||\theta||_2^2 + \bar{\gamma}||\phi||_2^2\right], \quad (6)$$

where we restrict $\rho, \gamma \in (0, 1)$ and define $\bar{\rho} := 1 - \rho$ and $\bar{\gamma} := 1 - \gamma$. This parameterization is one-to-one with the $\alpha, \beta$ parameterization (with $\alpha = \gamma\bar{\rho}/\rho$ and $\beta = \bar{\gamma}\bar{\rho}/\rho$) and it can be shown that $\nabla_{\theta,\phi}\mathcal{L}_{\rho,\gamma} \propto \nabla_{\theta,\phi}\mathcal{L}_{\alpha,\beta}$, thus minimizing one objective is equivalent to minimizing the other. Because $\rho$ and $\gamma$ are bounded we are able to completely cover

---

[2]This is under the assumption that either the networks have an unpenalized bias term in the final layer *or* that the data is standardized to have zero mean and unit variance.

the space of regularization combinations by searching over $(0, 1)^2$ whereas in the $\alpha, \beta$ parameterization $\alpha, \beta \in \mathbb{R}_{>0}$ are unbounded. Now, $\rho$ determines the relative importance between the likelihood and the total regularization imposed on both networks. On the other hand, $\gamma$ weights the proportion of total regularization between the mean and precision networks. Here, $\rho = 1$ corresponds to the MLE objective while $\rho \to 0$ could be interpreted as converging to the mode of the prior in a Bayesian setting. Fixing $\gamma = 1$ leads to an unregularized precision function while choosing $\gamma = 0$ results in an unregularized mean function.

**Qualitative Description of Phases** Model solutions across the space of $\rho$ and $\gamma$ hyperparameters exhibit different traits and behaviors. Similar to physical systems, this can be described as a collection of typical states or *phases* that make up a *phase diagram* as a whole. We find that these phase diagrams are typically consistent in shape across datasets and methodologies. Fig. 1 shows an example phase diagram along with model fits coming from specific $(\rho, \gamma)$ pairings. We argue that there are five primary regions of interest and qualitatively characterize them as follows:

• Region $U_I$: Both the mean and precision functions are heavily regularized. The likelihood is so lowly weighted it is as if the model had not seen the data. Regardless of the $\gamma$-value, the likelihood plays a minor role in the objective. The mean and standard deviation functions are constant through zero and 1 (the values they were initialized to).

• Region $U_{II}$: The mean function is still heavily regularized and tends to be flat, underfitting the data as in Region $U_I$. Despite the constant mean function, the precision function can still accommodate the residuals.

• Region $O_I$: The mean is heavily overfit and the residuals and corresponding standard deviations essentially vanish. Increasing $\rho \to 1$ yields true MLE fits (right side of the

Table 1: FT Limiting Cases. We provide intuition for Prop. 1 and match the limits to the phase diagram regions in Fig. 1.

| Regularization | Outcome |
|---|---|
| $\rho \to 1, \gamma \in [0,1]$ | This is equivalent to MLE. Approaching $\rho = 1$, we observe overfit mean solutions (see $O_\mathrm{I}$ and $O_\mathrm{II}$ in Fig. 1) across all $\gamma$. In theory, at $\rho = 1$, there is a contradiction implying no solution should exist. |
| $\rho \to 0, \gamma \in (0,1)$ | The objective is dominated by the regularizers—the data is completely ignored. This corresponds with region $U_\mathrm{I}$. In theory, the optimal solution at $\rho = 0$ is for both $\hat{\mu}, \hat{\Lambda}$ to be constant (flat) functions. |
| $\rho \in (0,1), \gamma \to 1$ | All regularization is placed on the mean function, leading to underfit mean. However, the precision is unregularized and the residuals are perfectly matched. This is the top row of the phase diagrams. |
| $\rho \in (0,1), \gamma \to 0$ | The mean is unregularized and the precision is strongly regularized. These fits are characterized by severe overfitting and can be found along the bottom row of the phase diagrams. |

phase diagram). This portion of the phase exists across a wide range of $\gamma$-values. Low values of $\gamma$ restrict the flexibility of the precision function, but due to the overfitting in the mean, the flexibility is not needed to fit the residuals.

• Region $O_\mathrm{II}$: The mean function does not overfit due to regularization, leaving large residuals for the lowly regularized precision function to overfit onto. The predicted standard deviation matches each residual perfectly.

• Region $S$: The mean and precision functions adapt to the data without overfitting. We conjecture that solutions in this region will provide the best generalization.

## 3 THEORETIC CONSIDERATIONS

We proceed to develop a theoretical description of the interplay between regularization strengths and resulting model behavior that captures the limiting behavior of heteroskedastic neural networks in the completely overparameterized regime. This tool allows us to analytically study edge cases of combinations of regularization strengths and find necessary conditions any pair of optimal mean and standard deviation functions must satisfy, agnostic of any specific model architecture. Furthermore, numerical solutions to our *field theory*, explained below, show good qualitative agreement with practical neural network implementations.

**Field Theory** Having discussed the effects of regularization on a heteroskedastic model on a qualitative level, we ask the following questions: *How much do these effects depend on any particular neural network architecture? Can we describe some of these effects on the function level, i.e., without resorting to neural networks?* To address these questions, we will establish *field theories* from statistical mechanics.

Field theories are statistical descriptions of random functions, rather than discrete or continuous random variables [Altland and Simons, 2010]. A *field* is a function assigning spatial coordinates to scalar values or vectors. Examples of physical fields are electric charge densities,

water surfaces, or vector fields such as magnetic fields. Low-energy configurations of fields can display recurring patterns (e.g., waves) or undergo phase transitions (e.g., magnetism) upon varying model parameters. Since we can think of a function as an infinite-dimensional vector, field theory requires the usage of *functional analysis* over plain calculus. For example, we frequently ask for the field that minimizes a free energy functional that we obtain by calculating a functional derivative that we set to zero. The advantage to moving to a function-space description is that all details about neural architectures are abstracted away as long as the neural network is sufficiently over-parameterized.

Firstly, we propose abstracting the neural networks $\hat{\mu}_\theta$ and $\hat{\Lambda}_\phi$ with nonparametric, twice-differentiable functions $\hat{\mu}$ and $\hat{\Lambda}$ respectively. Since these functions no longer depend on parameters, we cannot use $L_2$ penalties. A somewhat comparable substitute is to directly penalize the output "complexity" of the models, which can be measured via the *Dirichlet energy*: $\alpha \int p(x) ||\nabla \hat{\mu}(x)||_2^2 dx$ and $\beta \int p(x) ||\nabla \hat{\Lambda}(x)||_2^2 dx$. Note that these specific penalizations induce similar limiting behaviors for resulting solutions (i.e., $\alpha, \beta \to 0$ implies overfitting while $\to \infty$ implies constant functions). In the case where $\hat{\mu}_\theta$ and $\hat{\Lambda}_\phi$ are linear models, this gradient penalty is equivalent to an $L_2$ penalty. Further, networks trained with an $L_2$ weight regularization have empirically been found to have lower *geometric complexity*, a variant of *Dirichlet energy* [Dherin et al., 2022]. We also implement neural networks with *geometric complexity* regularization and present those results in Appendix E.

Using the assumptions outlined above and the same reparameterization of $(\alpha, \beta)$ to $(\rho, \gamma)$ as with the neural networks, the cross-entropy objective can be interpreted as an action functional of a corresponding two-dimensional FT,

$$\mathcal{L}_{\rho, \gamma}(\hat{\mu}, \hat{\Lambda}) = \int_{\mathcal{X}} p(x) \rho \int_{\mathcal{Y}} p(y \mid x) \log \hat{p}(y \mid x) dy \qquad (7)$$
$$+ p(x) \bar{\rho} \left[ \gamma ||\nabla \hat{\mu}(x)||_2^2 + \bar{\gamma} ||\nabla \hat{\Lambda}(x)||_2^2 \right] dx,$$

where $\hat{p}(y \mid x) = \mathcal{N}(y \mid \hat{\mu}(x), \hat{\Lambda}(x))$. This description as-

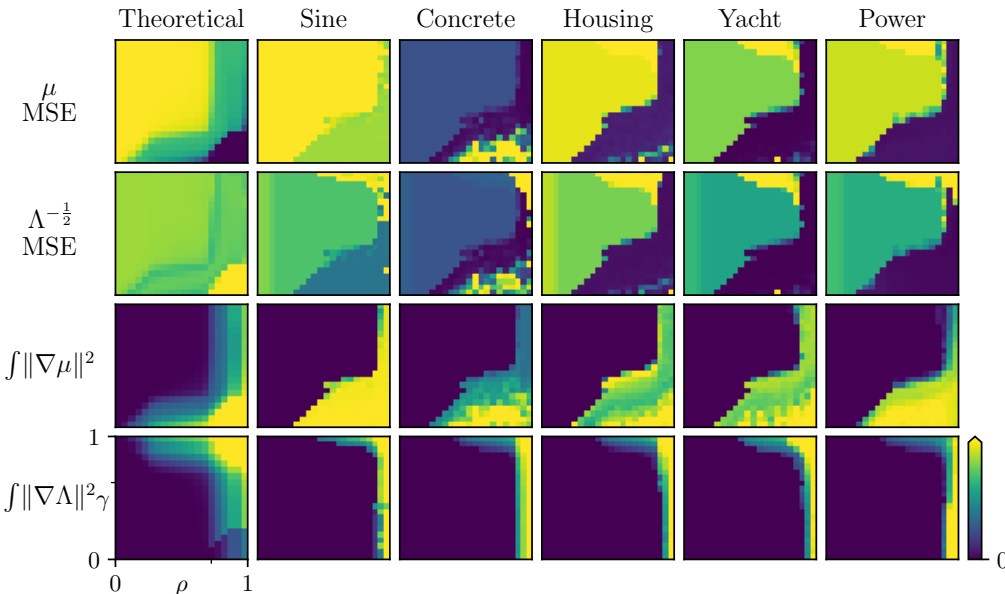

Figure 2: Array plot of metrics (rows) across different data or fitting techniques (columns). Leftmost column: results from our field theory (FT); remaining columns: results from fitting neural networks to data (data sets refer to test splits). Averaged results of six runs are shown. Intermediate ticks mark $\gamma = 0.5$ and $\rho = 0.5$ on the lower-left plot. Our FT aligns qualitatively well with empirical phase diagrams, with consistent phase transitions across models and datasets.

sumes a continuous data density $p(x)$, a continuous distribution over regression noise $p(y\,|\,x)$, and continuous functions $\hat{\mu}(x)$ and $\hat{\Lambda}(x)$ whose behavior we would like to study as a function of varying the regularizers $\rho$ and $\gamma$.

One can view the indexed set $y(\cdot) = \{y(x)\}_{x\in\mathcal{X}}$ as a stochastic process (specifically a white noise process scaled by true precision $\Lambda(x)$ and shifted by true mean $\mu(x)$). We are interested in the statistical properties of the field theory for any given realization of this stochastic process, $y(x)$, and ideally, we would average over multiple draws. However, for computational convenience, we restrict our attention to a single sample. This simplification is equivalent to considering a specific dataset and similar in spirit to fitting a heteroskedastic model to real data. This approximation yields the following simplified FT,

$$
\mathcal{L}_{\rho,\gamma}(\hat{\mu}, \hat{\Lambda}) \approx \int_{\mathcal{X}} p(x)\rho\left[\frac{1}{2}\hat{\Lambda}(x)\hat{r}(x)^2 - \frac{1}{2}\log\hat{\Lambda}(x)\right] \quad (8)
$$
$$
+ p(x)\bar{\rho}\left[\gamma||\nabla\hat{\mu}(x)||_2^2 + \bar{\gamma}||\nabla\hat{\Lambda}(x)||_2^2\right] dx,
$$

where $\hat{r}(x) := \hat{\mu}(x) - y(x)$. We are primarily interested in solutions $\hat{\mu}^*$ and $\hat{\Lambda}^*$ that minimize the FT $\mathcal{L}_{\rho,\gamma}(\hat{\mu}, \hat{\Lambda})$ as these are roughly analogous to models $\hat{\mu}_\theta$ and $\hat{\Lambda}_\phi$ that minimize penalized cross-entropy $\mathcal{L}_{\rho,\gamma}(\theta, \phi)$. We can gain insights into these solutions by taking functional derivatives of the FT with respect to $\hat{\mu}$ and $\hat{\Lambda}$ and setting them to zero.

The result of this procedure are stationary conditions in the form of *partial differential equations* for $\hat{\mu}^*$ and $\hat{\Lambda}^*$:

$$
\hat{\Lambda}^*(x)\hat{r}^*(x) = 2\frac{\bar{\rho}}{\rho}\gamma\frac{\Delta\hat{\mu}^*(x)}{p(x)}
$$
$$
\text{and} \quad \hat{r}^*(x)^2 = \frac{1}{\hat{\Lambda}^*(x)} + 4\frac{\bar{\rho}}{\rho}\bar{\gamma}\frac{\Delta\hat{\Lambda}^*(x)}{p(x)}, \quad (9)
$$

where $\hat{r}^*(x) = \hat{\mu}^*(x) - y(x)$ and $\Delta$ is the Laplace operator [Engel and Dreizler, 2011]. Note that these equalities hold true *almost everywhere* (a.e.) with respect to $p(x)$.

Interestingly, both resulting relationships include a regularization coefficient divided by the density of $x$. Intuitively, while the functions as a whole have a global level of regularization dictated by $\rho$ or $\gamma$, locally this regularization strength is augmented proportional to how likely the input is. This means that areas of high density in $x$ permit more complexity, while less likely regions are constrained to produce simpler outputs. Similarly, since $\Delta\hat{\mu}$ and $\Delta\hat{\Lambda}$ measure the *curvature* of these functions, we see that $\rho$ and $\gamma$ directly impact the complexity of $\hat{\Lambda}$ and $\rho$, as we expect.

**Numerically Solving the FT** Since the stationary conditions in Eq. (9) are too complex to be solved analytically, we discretize and minimize the FT to arrive at approximate solutions—in theory, we can do so to arbitrary precision. Let $\{x_i\}_{i=1}^{N_D}$ be a set of $D$ fixed points in $\mathcal{X}$ that we assume are evenly spaced. Define $\vec{\mu}, \vec{\Lambda}, \vec{y}$ to be $N_D$-dimensional vectors where for each $i$, $\vec{\mu}_i := \hat{\mu}(x_i), \vec{\Lambda}_i := \hat{\Lambda}(x_i), y_i := y(x_i)$. We solve for the optimal $\vec{\mu}$ and $\vec{\Lambda}$ using the discretized

approximation to Eq. (8) via gradient based optimization methods:

$$\mathcal{L}_{\rho,\gamma}(\vec{\mu}, \vec{\Lambda}) \approx \sum_{i=1}^{N_D} \rho \left[ \frac{1}{2} \vec{\Lambda}_i \left( y_i - \vec{\mu}_i \right)^2 - \frac{1}{2} \log \vec{\Lambda}_i \right]$$
$$+ \bar{\rho} \left[ \gamma ||\nabla \vec{\mu}_i||_2^2 + \bar{\gamma} ||\nabla \vec{\Lambda}_i||_2^2 \right], \quad (10)$$

and numerically approximate the gradients of $\hat{\mu}, \hat{\Lambda}$ by finite-difference methods [Fornberg, 1988].

**FT Insights**   The pair of constraints in Eq. (9) allow us to glean useful insights into the resulting regularized solutions by looking at edge cases of specific combinations of $\rho$ and $\gamma$ values. We summarize the theoretical properties of the limiting cases of $\rho$ and $\gamma$ approaching extreme values in the proposition below and in Table 1. Please refer to Appendix A.2 for the proofs of these claims.

**Proposition 1.** *Under the assumptions of our FT (see above), the following properties hold: (i) in the absence of regularization ($\rho = 1$), there are no solutions to the FT; (ii) in the absence of data ($\rho = 0$), there is no unique solution to the FT; and (iii) in order for there to exist a solution to the FT there must be regularization on the mean function.*

These limiting cases match our intuition conveyed earlier that also apply to the neural network context. Furthermore, if we assume that there do exist valid solutions for $\gamma, \rho \in (0, 1)$, it follows that the solutions should either undergo sharp transitions or smooth cross-overs between the behaviors described in the limiting cases when varying the regularization strengths. Section 4 shows that, empirically, these phase diagrams resemble Fig. 1. We leave the analytical justification for the types of boundaries and their shapes and placement in the phase diagram for future work.

# 4   EXPERIMENTS

The main focus of our experiments is to visualize the phase transitions in two-dimensional phase diagrams and identify summary statistics ("observables") that display them. We establish that these properties are independent of any particular neural network architecture by showing qualitative agreement with the field theory. Finally, through this exploratory analysis we discovered a method for finding well-suited combinations of $(\rho, \gamma)$-regularization strengths that reduces a two-dimensional hyperparameter search to one-dimension, allowing for the efficient identification of heteroskedastic model fits that neither over- nor underfit.

**Modeling Choices**   We chose $\hat{\mu}_\theta, \hat{\Lambda}_\phi$ to be fully-connected networks with three hidden layers of 128 nodes and leaky ReLU activation functions. The first half of training was only spent on fitting $\hat{\mu}_\theta$, while in the second half of training,

both $\hat{\mu}_\theta$ and $\hat{\Lambda}_\phi$ were jointly learned. This improves stability, since the precision is a dependent on the mean $\hat{\mu}_\theta$, and is similar in spirit to ideas presented in Skafte et al. [2019]. Complete training details can be found in Appendix B.2.

**Datasets**   We analyze the effects of regularization on several one-dimensional simulated datasets, standardized versions of the *Concrete* [Yeh, 2007], *Housing* [Harrison and Rubinfeld, 1978], *Power* [Tüfekci, 2014], and *Yacht* [Gerritsma, 1981] regression datasets from the UC Irvine Machine Learning Repository [Kelly et al.], and a scalar quantity from the ClimSim dataset [Yu et al., 2023]. We fit neural networks to the simulated and real-world data and additionally solve our FT for the simulated data. Detailed descriptions of the data are included in Appendix B.1. We present the results for a simulated sinusoidal (*Sine*) dataset as well as the four UCI regression datasets and have results for the other simulated datasets in Appendix B.5.

## 4.1   QUALITATIVE ANALYSIS

Our qualitative analysis aims at understanding architecture-independent aspects of heteroskedastic regression upon varying the regularization strength on the mean and variance functions, resulting in the observation of phase transitions.

**Metrics of Interest**   We are interested in how well-calibrated the resulting models are as well as how expressive the learned functions are. We compute two types of metrics on our experiments to summarize these properties. Firstly, we consider the mean squared error (MSE). We measure this quantity between predicted mean $\hat{\mu}_\theta(x_i)$ and target $y_i$, as well as between predicted standard deviation ($\hat{\Lambda}^{-1/2}(x_i)$) and absolute residual $|\hat{\mu}_\theta(x_i) - y_i|$. If the mean and standard deviation are well-fit to the data, both of these values should be low. We opt for $\Lambda^{-\frac{1}{2}}$ MSE due to its similarities to variance calibration [Skafte et al., 2019] and expected normalized calibration error [Levi et al., 2022]. Secondly, we evaluate the Dirichlet energy for the FT and its discrete analogue, geometric complexity [Dherin et al., 2022], for neural networks of the learned $\hat{\mu}_\theta, \hat{\Lambda}_\phi, \vec{\mu}, \vec{\Lambda}$. As previously mentioned, the Dirichlet energy of a function $f$ is defined as $\int_{\mathcal{X}} p(x) ||\nabla f(x)||_2^2 \, dx$. Meanwhile, geometric complexity is $N^{-1} \sum_{i=1}^{N} ||\nabla f(x)||_2^2$. Each quantity captures how expressive a learned function is, with more expressive functions yielding a higher value and is analogous (or equivalent) to the quantity we penalize in the FT setting.

**Plot Interpretation**   We present summaries of the fitted models in grids with $\rho$ on the $x$-axis and $\gamma$ on the $y$-axis in Fig. 2. The far right column ($\gamma = 1$) corresponds to MLE solutions. The main focus is on qualitative traits of fits under different levels of regularization and how they behave in a relative sense, rather than a focus on absolute values. Fig. 3 show the summary statistics along the slice where

Table 2: Comparison of a deep heteroskedastic regression model with diagonal regularization search with $\beta$-NLL [Seitzer et al., 2022] and two conformal prediction implementations. For details on the selection criteria of the heteroskedastic model see Appendix D.2. The final two columns are comparisons against models selected in the same way as in our suggestion, but trained on half of the data in a split conformal fashion. The third column has uniform bandwidth (homoskedastic) assumptions while the fourth column has a locally adaptive [Lei et al., 2018] (heteroskedastic) bandwidth. Possibly due to the reduced training size, performance suffers. In a conformal setting the "standard deviation" does not have an obvious analogue. We calibrate the bandwidths to be set to the 0.682 quantile because $\pm 1$ standard deviation covers $\approx 68.2\%$ of a standard normal distribution. Lowest mean value of for each quantity is bolded. We report the average and standard deviations of $\mu$- and $\Lambda^{-\frac{1}{2}}$-MSE across six runs on test data.

| Dataset | Metric | Heteroskedastic | $\beta$-NLL | Conformal | Conformal (local) |
|---|---|---|---|---|---|
| Sine | $\mu$ MSE | $0.80 \pm 0.00$ | $0.69 \pm 0.05$ | $\mathbf{0.54} \pm 0.09$ | $0.82 \pm 0.00$ |
| | $\Lambda^{-\frac{1}{2}}$ MSE | $0.80 \pm 0.00$ | $0.52 \pm 0.07$ | $0.36 \pm 0.13$ | $\mathbf{0.33} \pm 0.00$ |
| Concrete | $\mu$ MSE | $\mathbf{0.11} \pm 0.02$ | $0.55 \pm 0.30$ | $0.27 \pm 0.01$ | $0.80 \pm 0.00$ |
| | $\Lambda^{-\frac{1}{2}}$ MSE | $0.30 \pm 0.51$ | $1.09 \pm 0.20$ | $\mathbf{0.09} \pm 0.00$ | $0.43 \pm 0.00$ |
| Housing | $\mu$ MSE | $1.22 \pm 0.00$ | $0.32 \pm 0.05$ | $\mathbf{0.31} \pm 0.00$ | $\mathbf{0.31} \pm 0.00$ |
| | $\Lambda^{-\frac{1}{2}}$ MSE | $0.76 \pm 0.00$ | $0.88 \pm 0.03$ | $\mathbf{0.13} \pm 0.00$ | $0.14 \pm 0.00$ |
| Power | $\mu$ MSE | $\mathbf{0.04} \pm 0.01$ | $0.09 \pm 0.01$ | $0.18 \pm 0.00$ | $0.19 \pm 0.00$ |
| | $\Lambda^{-\frac{1}{2}}$ MSE | $\mathbf{0.03} \pm 0.01$ | $0.31 \pm 0.37$ | $\mathbf{0.03} \pm 0.00$ | $\mathbf{0.03} \pm 0.00$ |
| Yacht | $\mu$ MSE | $\mathbf{0.01} \pm 0.01$ | $\mathbf{0.01} \pm 0.01$ | $0.04 \pm 0.00$ | $0.84 \pm 0.00$ |
| | $\Lambda^{-\frac{1}{2}}$ MSE | $\mathbf{0.01} \pm 0.01$ | $1.33 \pm 0.02$ | $\mathbf{0.01} \pm 0.00$ | $0.49 \pm 0.00$ |
| Solar Flux | $\mu$ MSE | $0.29 \pm 0.00$ | $0.38 \pm 0.00$ | $\mathbf{0.05} \pm 0.00$ | $0.33 \pm 0.00$ |
| | $\Lambda^{-\frac{1}{2}}$ MSE | $0.12 \pm 0.00$ | $0.32 \pm 0.00$ | $\mathbf{0.01} \pm 0.00$ | $0.37 \pm 0.01$ |

$\rho = 1 - \gamma$. Zero on these plots corresponds to the upper left corner while one corresponds to the lower right corner. We provide model fits arranged in grids of the same orientation for the field theory and neural networks on the *Sine* dataset in Figs. 10 and 11.

**Observation 1:** *Our metrics show sharp phase transitions upon varying $\rho, \gamma$, as in a physical system.*

Fig. 2 and Fig. 3 show a sharp transition, both leading to worsening and improving performance when moving along the minor diagonal. In totality, across all metrics, the five regions are apparent. But not all of the regions in Fig. 1 appear in the heatmaps of each metric. For example, region $O_{\text{II}}$ does not always appear in the metrics related to the mean. When using neural networks to approximate $\mu$ and $\Lambda$, there are sharper boundaries between phases than in the FT's numerical solutions. The boundary between $U_{\text{II}}$ and $O_{\text{I}}$ is sharply observed in the plots of $\int ||\nabla \mu(x)||_2^2 \, dx$. However, in terms of $\mu$ MSE, a smoother transition (i.e., region $S$) is visible.

**Observation 2:** *The FT insights and observed phases are consistent with the numerically solved FT and the results from fitting neural networks. Thus, our results are not tied to a specific architecture or dataset.*

In alignment with our theoretical insights, phases $U_{\text{I}}$ and $O_{\text{I}}$ exhibit consistent behavior across $\gamma$-values (vertical slices in Fig. 2). Qualitatively, we find the same types of phase diagrams and phase transitions across all considered data

sets. Empirically, we observe that boundaries between regions of interest are similar in shape across datasets but not quantitatively the same, i.e., phase transitions occur at differing levels of regularization for different data sets of different dimensionality.

In the right-hand columns ($\rho \to 1$), there is near-perfect matching of the data by the mean function and this is also visible in the lower rows ($\gamma \to 0$). Within the metrics we assess, the shapes of the regions vary with regularization strength in a similar fashion on all datasets. In the plots of $\int ||\nabla \Lambda(x)||_2^2 \, dx$, the region where $\Lambda$ is flatter covers a larger area compared to the phase diagram showing $\int ||\nabla \mu(x)||_2^2 \, dx$. That is, for the same proportion of regularization as the mean, the precision remains flatter.

### 4.2 QUANTITATIVE ANALYSIS

Our quantitative analysis aims to demonstrate the practical implications of our qualitative investigations that result in better calibration properties.

**Observation 3:** *We can search along $\rho = 1 - \gamma$ to find a well-calibrated $(\rho, \gamma)$-pair from region $S$.*

Our FT indicates that a slice across the minor diagonal of the phase diagram should always cross the $S$ region (see Fig. 1). Fig. 3 show that by searching along this diagonal, we indeed find a combination of regularization strengths where both $\hat{\mu}_\theta$ and $\hat{\Lambda}_\phi$ generalize well to held-out test data.

This implies that there is no need to search all of the two-dimensional space, but only a single slice which reduces the number of models to fit from $O(N^2)$ to $O(N)$, where $N$ is the number of $\rho$ and $\gamma$ values that are tested.

Fig. 3 shows that along the minor diagonal the performance is initially poor, improves, and then drops off again. These shifts from strong to weak performance are sharp. The regularization pairings that result in optimal performance with respect to $\mu$- and $\Lambda^{-1/2}$-MSE are near each other along this diagonal for the real-world test data. As the theory predicts, the performance becomes highly variable as we approach the MLE solutions and the FT fails to converge in this region. In practice, we propose searching along this line to find the $(\rho, \gamma)$-combinations that minimize the $\mu$- and $\Lambda^{-\frac{1}{2}}$-MSEs and averaging the regularization strengths to fit a model. We compare models chosen by our diagonal line search to two heteroskedastic modeling baselines in Appendix D on the synthetic and UCI datasets as well as a scalar quantity from the ClimSim dataset [Yu et al., 2023]. We present a subset of the results below in Table 2. In most cases the model chosen via the diagonal line search was competitive or better than the baselines.

## 5 RELATED WORK

Uncertainty can be divided into epistemic (model) and aleatoric (data) uncertainty [Hüllermeier and Waegeman, 2021], the latter of which can be further divided into homoskedastic (constant over input space) and heteroskedastic (varies over input space). Handling heteroskedastic noise historically has been and continues to be an active area of research in statistics [Huber, 1967, Eubank and Thomas, 1993, Le et al., 2005, Uyanto, 2022] and machine learning [Abdar et al., 2021], but is less common in deep learning [Kendall and Gal, 2017, Fortuin et al., 2022], probably due to pathologies that we analyze in this work. Heteroskedastic noise modeling can be interpreted as reweighting the importance of datapoints during training time, which Wang et al. [2017] and Mandt et al. [2016] show to be beneficial in the presence of corrupted data and Khosla et al. [2022] in active learning.

To the best of our knowledge, Nix and Weigend [1994] were the first to model a mean and standard deviation function with neural networks and Gaussian likelihood. Skafte et al. [2019] suggest changing the optimization loop to train the mean and standard deviation networks separately, treating the standard deviations variationally and integrating them out as Takahashi et al. [2018] does in the context of VAEs, accounting for the location of the data when sampling, and setting a predefined global variance when extrapolating. Stirn and Knowles [2020] also perform amortized VI on the standard deviations and evaluate their model from the perspective of posterior predictive checks. Seitzer et al. [2022] provide an in-depth analysis of the shortcomings of MLE es-

timation in this setting and adjust the gradients during training to avoid pathological behavior. Stirn et al. [2023] extend the idea of splitting mean and standard deviation network training in a setting where there are several shared layers to learn a representation before emitting mean and standard deviation. Finally, Immer et al. [2023] take a Bayesian approach to the problem and use Laplace approximation on the marginal likelihood to perform empirical Bayes. This allows for regularization to be applied through the prior and for separation of model and data uncertainty. While these works propose practical solutions, in contrast to our work, none of them study the theoretical underpinnings of these pathologies, let alone in a model- or data-agnostic way.

## 6 CONCLUSION

We have used field-theoretical tools from statistical physics to derive a nonparametric free energy, which allowed us to produce analytical insights into the pathologies of deep heteroskedastic regression. These insights generalize across models and datasets and provide a theoretical explanation for the need for carefully tuned regularization in these models, due to the presence of sharp phase transitions between pathological solutions.

We have also presented a numerical approximation to this theory, which empirically agrees with neural network solutions to synthetic and real-world data. Insights from the theory have informed a method to tune the regularization to arrive at well-calibrated models more efficiently than would naïvely be the case. Finally, we hope that this work will open an avenue of research for using ideas from theoretical physics to study the collective effects and nonlinear phenomena frequently encountered in large-scale deep learning models [Bamler and Mandt, 2018].

**Limitations** Our FT and subsequent analysis are restricted to regression problems. From an uncertainty quantification perspective, the models we discuss only account for the aleatoric uncertainty. Though our use of regularizers has a Bayesian interpretation, we are not performing Bayesian inference and do not account for epistemic uncertainty [Papamarkou et al., 2024]. Solving the FT under a fully Bayesian framework would result in stochastic PDE solutions. We leave analysis of this setting to future work. Additionally, our suggestion to search $\rho = 1 - \gamma$ to find good hyperparameter settings appears to be valid, but requires fitting many models. Ideally, one might hope to use the field theory directly to find optimal regularization settings for real-world models, but our numerical approach is currently not accurate enough for this use case.

**Acknowledgements** Eliot Wong-Toi acknowledges support from the Hasso Plattner Research School at UC Irvine. Alex Boyd acknowledges support from the National Sci-

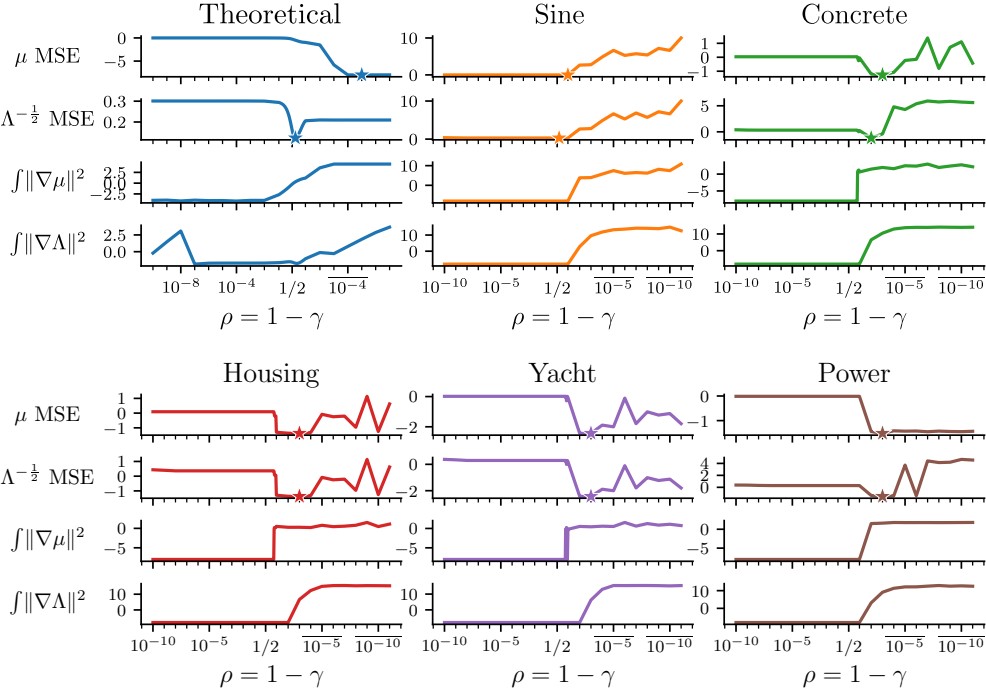

Figure 3: Test metrics for six runs achieved along the $\rho = 1 - \gamma$ minor diagonal. Stars indicate minimum MSE values. All metrics are reported on a $\log_{10}$ scale. $\rho$ values are shown on a logit scale with $\overline{10^k} := 1 - 10^k$. From left to right, note the sharp decrease in test metric values, especially in the solutions to neural network models followed by a typical smoother increase. This empirically supports the existence of the well-calibrated $S$ phase shown in Fig. 1.

ence Foundation Graduate Research Fellowship grant DGE-1839285. Vincent Fortuin was supported by a Branco Weiss Fellowship. Stephan Mandt acknowledges support by the IARPA WRIVA program, the National Science Foundation (NSF) under the NSF CAREER Award 2047418; NSF Grants 2003237 and 2007719, the Department of Energy, Office of Science under grant DE-SC0022331, as well as gifts from Intel, Disney, and Qualcomm.

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

# Understanding Pathologies of Deep Heteroskedastic Regression (Supplementary Material)

**Eliot Wong-Toi**[1]      **Alex Boyd**[1]      **Vincent Fortuin**[2]      **Stephan Mandt**[1,3]

[1]Department of Statistics, University of California, Irvine, USA
[2]Helmholtz AI, Munich, Germany
[3]Department of Computer Science, University of California, Irvine, USA

## A  THEORETICAL DETAILS

### A.1  FULL FUNCTIONAL DERIVATIVES

Our FT is:

$$\mathcal{L}_{\rho,\gamma}(\hat{\mu}, \hat{\Lambda}) \approx \int_{\mathcal{X}} p(x)\rho\left[\frac{1}{2}\hat{\Lambda}(x)\hat{r}(x)^2 - \frac{1}{2}\log\hat{\Lambda}(x)\right] + p(x)\bar{\rho}\left[\gamma||\nabla\hat{\mu}(x)||_2^2 + \bar{\gamma}||\nabla\hat{\Lambda}(x)||_2^2\right] dx$$

and its functional derivatives are

$$\begin{cases} \frac{\delta\mathcal{L}}{\delta\hat{\mu}} &= p(x)\rho\hat{\Lambda}(x)\hat{r}(x) - 2\bar{\rho}\gamma\Delta\hat{\mu}(x) \\ \frac{\delta\mathcal{L}}{\delta\hat{\Lambda}} &= \frac{p(x)\rho}{2}\left[\hat{r}(x)^2 - \frac{1}{\hat{\Lambda}(x)}\right] - 2\bar{\rho}\bar{\gamma}\Delta\hat{\Lambda}(x), \end{cases} \tag{11}$$

where $\hat{r}(x) = y(x) - \hat{\mu}(x)$ After setting equal to zero we arrive at

$$\begin{cases} \hat{\Lambda}^*(x)(\hat{\mu}^*(x) - y(x)) = 2\frac{\bar{\rho}}{\rho}\gamma\frac{\Delta\hat{\mu}^*(x)}{p(x)} \\ (y(x) - \hat{\mu}^*(x))^2 = \frac{1}{\hat{\Lambda}^*(x)} + 4\frac{\bar{\rho}}{\rho}\bar{\gamma}\frac{\Delta\hat{\Lambda}^*(x)}{p(x)}. \end{cases} \tag{12}$$

### A.2  PROOFS

**Proposition 1.** *Assuming there exists twice differentiable functions $\mu : \mathbb{R}^d \to \mathbb{R}, \Lambda : \mathbb{R}^d \to \mathbb{R}_{>0}$, the following properties hold*

   *i  In the absence of regularization ($\rho = 1$), there are no solutions to the FT.*

  *ii  In the absence of data ($\rho = 0$), there is no unique solution to the FT.*

 *iii  In order for there to exist a solution to the FT there must be regularization on the mean function.*

*Proof.* Without loss of generality, we consider a uniform $p(x)$ and drop it from the equations.

(i) When $\rho = 1$ the necessary conditions for an optima are

$$\begin{cases} \hat{\Lambda}^*(x)(\hat{\mu}^*(x) - y(x)) = 0 \\ (\hat{\mu}^*(x) - y(x))^2 = \frac{1}{\hat{\Lambda}^*(x)} \end{cases} \tag{13}$$

$$\implies \begin{cases} \hat{\Lambda}^*(x)(\hat{\mu}^*(x) - y(x)) = 0 \\ \hat{\Lambda}^*(x)(\hat{\mu}^*(x) - y(x))^2 = 1 \end{cases} \tag{14}$$

$$\implies \begin{cases} \hat{\Lambda}^*(x)(\hat{\mu}^*(x) - y(x)) = 0 \\ 0 \times (\hat{\mu}^*(x) - y(x)) = 1 \end{cases} \tag{15}$$

$$\implies 0 = 1 \tag{16}$$

which is a contradiction and there cannot exist $\mu, \Lambda$ that are solutions.

(ii) When $\rho = 0$ the integral we seek to maximize is:

$$\mathcal{L}_{\rho,\gamma}(\hat{\mu}, \hat{\Lambda}) = \int_{\mathcal{X}} \rho \int_{\mathcal{Y}} p(y \,|\, x) \log \hat{p}(y \,|\, x) dy + \bar{\rho} \left[ \gamma || \nabla \hat{\mu}(x) ||_2^2 + \bar{\gamma} || \nabla \hat{\Lambda}(x) ||_2^2 \right] dx \tag{17}$$

$$= \int_{\mathcal{X}} \left[ \gamma || \nabla \hat{\mu}(x) ||_2^2 + \bar{\gamma} || \nabla \hat{\Lambda}(x) ||_2^2 \right] dx \tag{18}$$

where we $\hat{p}(y|x) = \mathcal{N}(y \,|\, \hat{\mu}(x), \hat{\Lambda}(x))$. Each term in this integral is non-negative, so the minimum value it could be is zero. Any pair of constant functions $\mu, \Lambda$ will minimize this integral, of which there are infinitely many.

(iii) In the $(\alpha, \beta)$-regularization, it is equivalent to say $\alpha > 0$ is a necessary condition for there to exist a solution to the FT. Recall that we seek to minimize

$$\mathcal{L}_{\alpha,\beta}(\hat{\mu}, \hat{\Lambda}) = \int_{\mathcal{X}} p(x)(-\log \hat{p}(y|x) + \alpha || \nabla \hat{\mu}(x) ||_2^2 + \beta || \nabla \hat{\Lambda}(x) ||_2^2) dx$$

where we $\hat{p}(y|x) = \mathcal{N}(y|\hat{\mu}(x), \hat{\Lambda}(x))$. Suppose $\alpha = 0$. Then the functional simplifies to

$$\min_{\hat{\mu}, \hat{\Lambda}} \mathcal{L}_{\alpha,\beta}(\hat{\mu}, \hat{\Lambda}) = \min_{\hat{\mu}, \hat{\Lambda}} \int_{\mathcal{X}} p(x)(-\log \hat{p}(y|x) + \beta || \nabla \hat{\Lambda}(x) ||_2^2) dx \tag{19}$$

$$\leq \min_{\hat{\mu}, \underline{\hat{\Lambda}}} \int_{\mathcal{X}} p(x) \frac{1}{2} (\underline{\hat{\Lambda}}(x) \hat{r}(x)^2 - \log \underline{\hat{\Lambda}}(x)) + p(x) \beta || \nabla \underline{\hat{\Lambda}}(x) ||_2^2 dx \tag{20}$$

$$= \min_{\hat{\mu}, \underline{\hat{\Lambda}}}(x) \int_{\mathcal{X}} p(x) \frac{1}{2} (\underline{\hat{\Lambda}}(x) \hat{r}(x)^2 - \log \underline{\hat{\Lambda}}(x)) dx \tag{21}$$

where $\underline{\hat{\Lambda}}$ is a constant function and $\hat{r}(x) = y(x) - \hat{\mu}(x)$. This provides an upper bound on the integral as we are looking at a restricted class of possible precision functions. Since the precision function is constant the gradient penalty, $|| \nabla \hat{\Lambda}(x) ||_2^2$, is zero. There is no penalty on $\hat{\mu}$ so it can perfectly pass through every data point and the contribution of $\underline{\hat{\Lambda}}(x)(\hat{\mu}(x) - y(x))^2$ is zero while $-\log \underline{\hat{\Lambda}}(x)$ can become arbitrarily negative. Thus there is no solution if $\alpha = 0$.

$\square$

# B EXPERIMENTAL DETAILS

## B.1 DATASETS

We chose 64 datapoints in each of the simulated datasets. The generating processes for each simulated dataset is included in Table 3 and can be seen in Fig. 4. The homoskedastic data is simulated in the same way, but with $f(x) = 1$. For testing, we simulate a new dataset of 64 datapoints with the same process. Table 4 summarizes the UCI datasets. We provide a description of the ClimSim climate data in Appendix D.4.

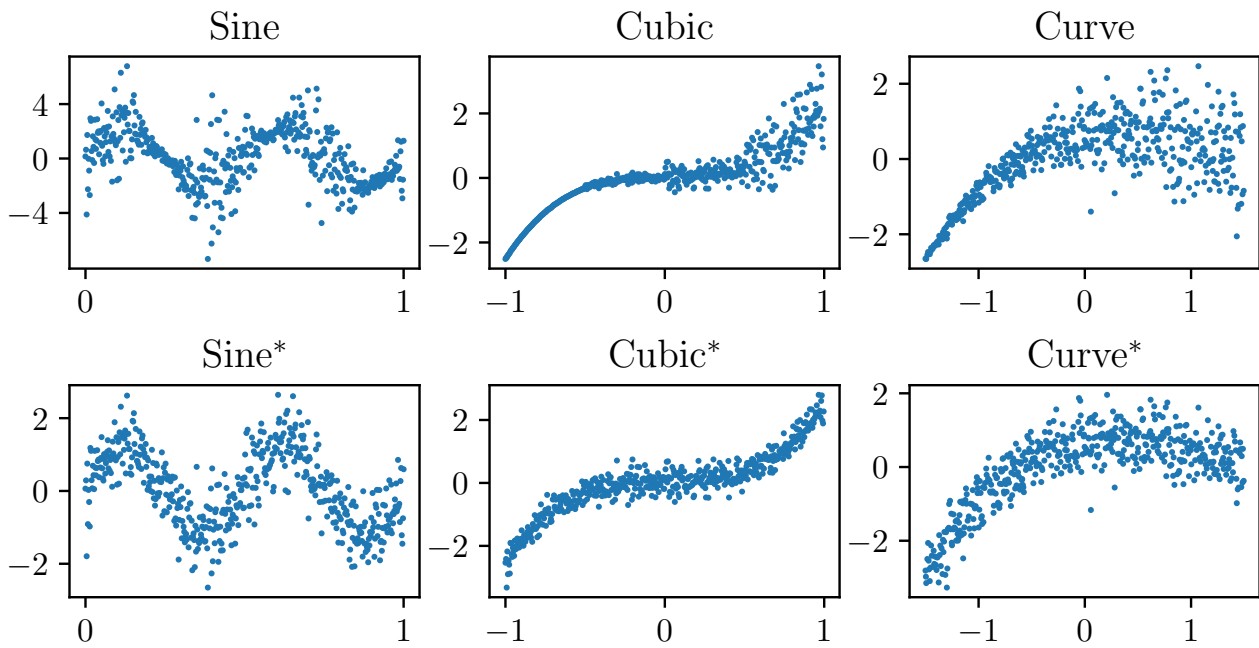

Figure 4: Visualization of heteroskedastic and homoskedastic versions of simulated datasets. Specific details for the functional form of these can be found in Table 3.

Table 3: Simulated datasets. Each dataset is defined by a true $\mu$ function and then a noise function $f$. All data is generated as $\mu(x) + \epsilon(x)$ where $\epsilon(x) \sim \mathcal{N}(0, f(x)^2)$. After the datasets were generated they were scaled to have mean zero and standard deviation one. The homoskedastic versions of each dataset fix $f(x) = 1$. The datasets are shown in Fig. 4.

| Dataset | Mean ($\mu$) | Noise Pattern ($f$) | Domain |
|---------|--------------|---------------------|--------|
| Sine | $\mu(x) = 2\sin(4\pi x)$ | $f(x) = \sin(6\pi x) + 1.25$ | $x \in [0, 1]$ |
| Cubic | $\mu(x) = x^3$ | $f(x) = \begin{cases} 0.1 & \text{for } x < -0.5 \\ 1 & \text{for } x \in [-0.5, 0.0) \\ 3 & \text{for } x \in [0.0, 0.5) \\ 10 & \text{for } x \geq .5 \end{cases}$ | $x \in [-1, 1]$ |
| Curve | $\mu(x) = x - 2x^2 + 0.5x^3$ | $f(x) = x + 1.5$ | $x \in [-1.5, 1.5]$ |

Table 4: UCI dataset.

| Dataset | Train Size | Test Size | Input Dimension |
|---------|------------|-----------|-----------------|
| Concrete | 687 | 343 | 8 |
| Housing | 337 | 168 | 13 |
| Power | 6379 | 3189 | 4 |
| Yacht | 204 | 102 | 6 |

## B.2 TRAINING DETAILS

We take 22 values of $\gamma, \rho$ that range from $10^{-10}$ up to $1 - 10^{-5}$ on a logit scale for all of the experiments run on neural networks. The exact values were ($\rho, \gamma \in \{0.9999, 0.999, 0.99, 0.9, 0.8, 0.7, 0.6, 0.5, 0.4, 0.3, 0.2, 0.1, 0.01, 0.001, 0.0001,$ $0.00001, 0.000001, 0.0000001, 0.00000001, 0.000000001, 0.0000000001, 0.00000000001\}$). For the FTs we take 20 values from $10^{-6}$ up to $1 - 10^{-7}$ also on a logit scale ($\rho, \gamma \in \{0.999999, 0.99999, 0.99999, 0.9999, 0.999, 0.99, 0.9, 0.8, 0.7, 0.6,$ $0.5, 0.4, 0.3, 0.2, 0.1, 0.01, 0.001, 0.0001, 0.00001, 0.000001\}$). This scaling increases the absolute density of points evaluated near the extreme cases of 0 and 1 where the theoretical analysis of the FT focused. The ranges differ slightly due to numerical stability during the fitting. The limiting cases of $\gamma, \rho \in \{0, 1\}$ were omitted for numerical stability and the ranges of values for the FTs vs neural networks vary slightly for the same reason. The values of $\rho, \gamma$ that were taken along the $\rho = 1 - \gamma$ line were $\rho, \gamma \in \{0.0, 1.0 \times 10^{-11}, 1.0 \times 10^{-10}, 1.0 \times 10^{-9}, 1.0 \times 10^{-8}, 1.0 \times 10^{-7}, 1.0 \times 10^{-6}, 1.0 \times 10^{-5}, 1.0 \times 10^{-4}, 1.0 \times 10^{-3},$ $0.01, 0.1, 0.11, 0.12, 0.13, 0.14, 0.15, 0.16, 0.17, 0.18, 0.19, 0.20, 0.21, 0.22, 0.23, 0.24, 0.25, 0.26, 0.27, 0.28, 0.29, 0.30,$ $0.31, 0.32, 0.33, 0.34, 0.35, 0.36, 0.37, 0.38, 0.39, 0.40, 0.41, 0.42, 0.43, 0.44, 0.45, 0.46, 0.47, 0.48, 0.49, 0.50, 0.51, 0.52,$ $0.53, 0.54, 0.55, 0.56, 0.57, 0.58, 0.59, 0.60, 0.61, 0.62, 0.63, 0.64, 0.65, 0.66, 0.67, 0.68, 0.69, 0.70, 0.71, 0.72, 0.73,$ $0.74, 0.75, 0.76, 0.77, 0.78, 0.79, 0.80, 0.81, 0.82, 0.83, 0.84, 0.85, 0.86, 0.87, 0.88, 0.89, 0.9, 0.99, 0.999, 0.9999, 1.0\}$. All experiments were run on Nvidia Quadro RTX 8000 GPUs. Approximately 500 total GPU hours were used across all experiments.

## B.3 METRICS

- Geometric complexity: For the one-dimensional datasets the function is evaluated on a dense grid and then the gradients are approximated via finite differences and a trapezoidal approximation to the integral is taken. In the case of the FT, we only assess the function on the solved for, discretized points while with the neural networks we interpolate between points. For the higher-dimensional UCI datasets the gradients are also numerically approximated in the same way but only at the points in the train/test sets.

- MSE: In the fully non-parametric, unconstrained setting, the maximum likelihood estimates at each $x_i$ are $\hat{\mu}(x_i) = y(x_i)$ and $\hat{\Lambda}(x_i) = (y(x_i) - \mu(x_i))^{-2} \implies \hat{\Lambda}^{-1/2}(x_i) = |y(x_i) - \mu(x_i)|$, serving as motivation for checking these differences.

**Variability over runs** The experiments were each run six times with different seeds. The standard deviations over the metrics displayed in Fig. 2 are shown in Fig. 5. The Sobolev norms show that there is the most variability in the overfitting regions $O_\mathrm{I}$ and parts of $O_\mathrm{II}$. This indicates that the functions themselves vary across runs. However, when turning to quality of fits, the MSEs show a different pattern of regions of instability, and $O_\mathrm{I}$ has low variability in terms of actual performance.

## B.4 FIELD THEORY

For the discretized field theory we take $n_{ft} = 4096$ evenly spaced points on the interval $[-1, 1]$. There are two datapoints placed beyond $[-1, 1]$ because the method we use to estimate the gradients requires the datapoints to have left and right neighbors. These datapoints were not included when computing our metrics. Of these 4096 datapoints 64 were randomly selected to be used for training neural networks $\hat{\mu}_\theta, \hat{\Lambda}_\phi$. The field theory results were consistent across choices of $n_{ft} \in \{256, 512, 1024, 2048, 4096\}$. We present results for $n_{ft} = 4096$ in the main paper. We train for $100000$ epochs and use the Adam optimizer with a basic triangular cycle that scales initial amplitude by half each cycle on the learning rate. The minimum and maximum learning rates were 0.0005 and 0.01. The cycles were 5000 epochs long. We clip the gradients at 1000. A subset of the fits can be seen in Fig. 10.

## B.5 SIMULATED DATA WITH NEURAL NETWORKS

For all of the simulated datasets except for *Sine* we train for $600000$ epochs and use the Adam optimizer with a basic triangular cycle that scales initial amplitude by half each cycle on the learning rate. The minimum and maximum learning rates were 0.0001 and 0.01. The cycles were 50000 epochs. The first 250000 epochs are only spend on training $\hat{\mu}_\theta$ while the remaining 350000 epochs are spent training both $\hat{\mu}_\theta, \hat{\Lambda}_\phi$. We clip the gradients at 1000. The training for the *Sine* dataset was the same, except trained for 2500000 epochs. A subset of the fits for the *Sine* dataset can be seen in Fig. 11.

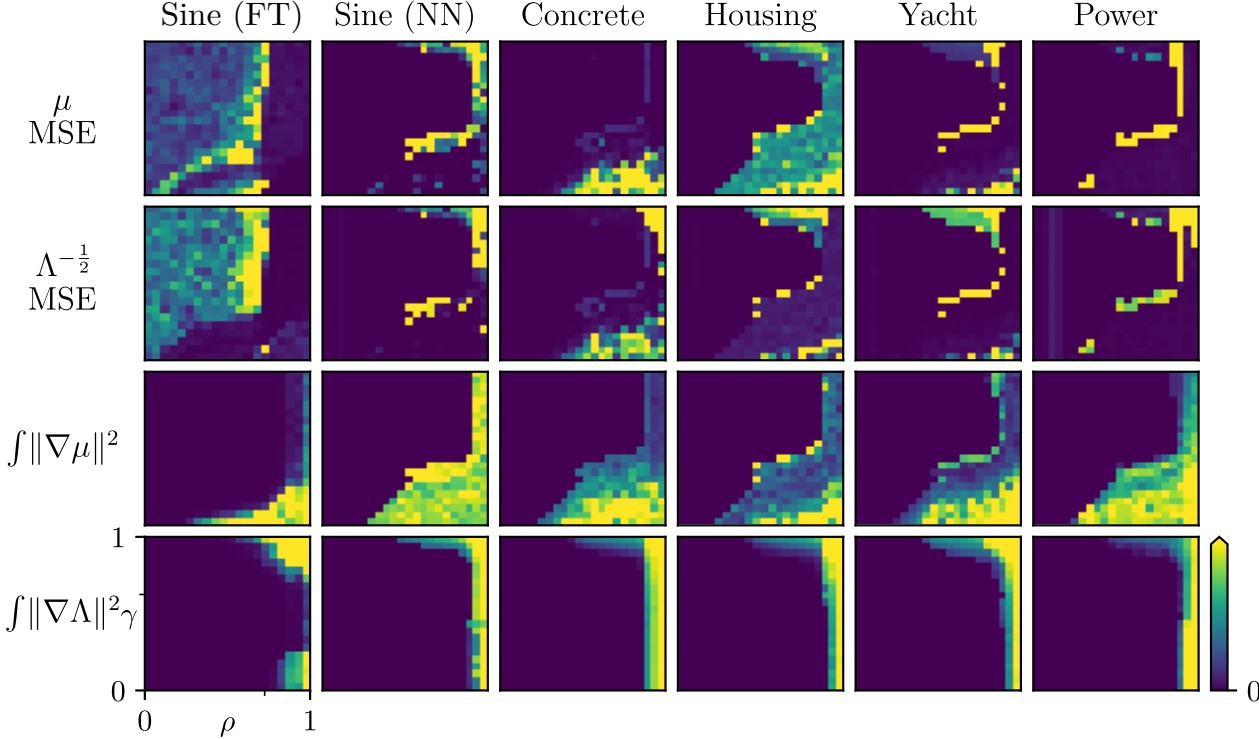

Figure 5: The standard deviation over the six runs of each metric shown in Fig. 2

## B.6 UCI DATA WITH NEURAL NETWORKS

For the *concrete, housing* and *yacht* datasets we train for 500000 epochs and use the Adam optimizer with a basic triangular cycle that scales initial amplitude by half each cycle on the learning rate. The minimum and maximum learning rates were 0.0001 and 0.01. The cycles were 50000 epochs. The first 250000 epochs are only spend on training $\hat{\mu}_\theta$ while the remaining 250000 epochs are spent training both $\hat{\mu}_\theta, \hat{\Lambda}_\phi$. Meanwhile on the *power* dataset, we had to use minibatching due to the size of the dataset. We used minibatches of 1000 and trained for 50000 total epochs with the first 25000 dedicated solely to $\hat{\mu}_\theta$ and the remainder training both $\hat{\mu}_\theta, \hat{\Lambda}_\phi$. The same cyclic learning rate was used but with cycle length 5000. We clip the gradients at 1000.

## B.7 PRACTICAL SUGGESTION

We can also view the $\rho = 1 - \gamma$ line that we search from the perspective of the $\alpha, \beta$ parameterization of the regularizers. Let $\rho, \gamma \in (0, 1)$ such that $\rho = 1 - \gamma$. Furthermore, we know that $\alpha = \frac{1-\rho}{\rho}\gamma$ and that $\beta = \frac{1-\rho}{\rho}(1 - \gamma)$. If we are interested in the model settings for $(\rho(t) = t, \gamma(t) = 1 - t)$ for $t \in (0, 1)$, it then follows that we are equivalently interested in

$$(\alpha(t), \beta(t)) = \left( \frac{1 - \rho(t)}{\rho(t)}\gamma(t), \frac{1 - \rho(t)}{\rho(t)}(1 - \gamma(t)) \right)$$
$$= \left( \frac{1 - t}{t}(1 - t), \frac{1 - t}{t}t \right)$$
$$= \left( \frac{(1 - t)^2}{t}, 1 - t \right)$$
$$\implies \begin{cases} t & = 1 - \beta(t) \\ \alpha(t) & = \frac{\beta(t)^2}{t} \end{cases} \text{ or } \sqrt{t\alpha(t)} = \beta(t).$$

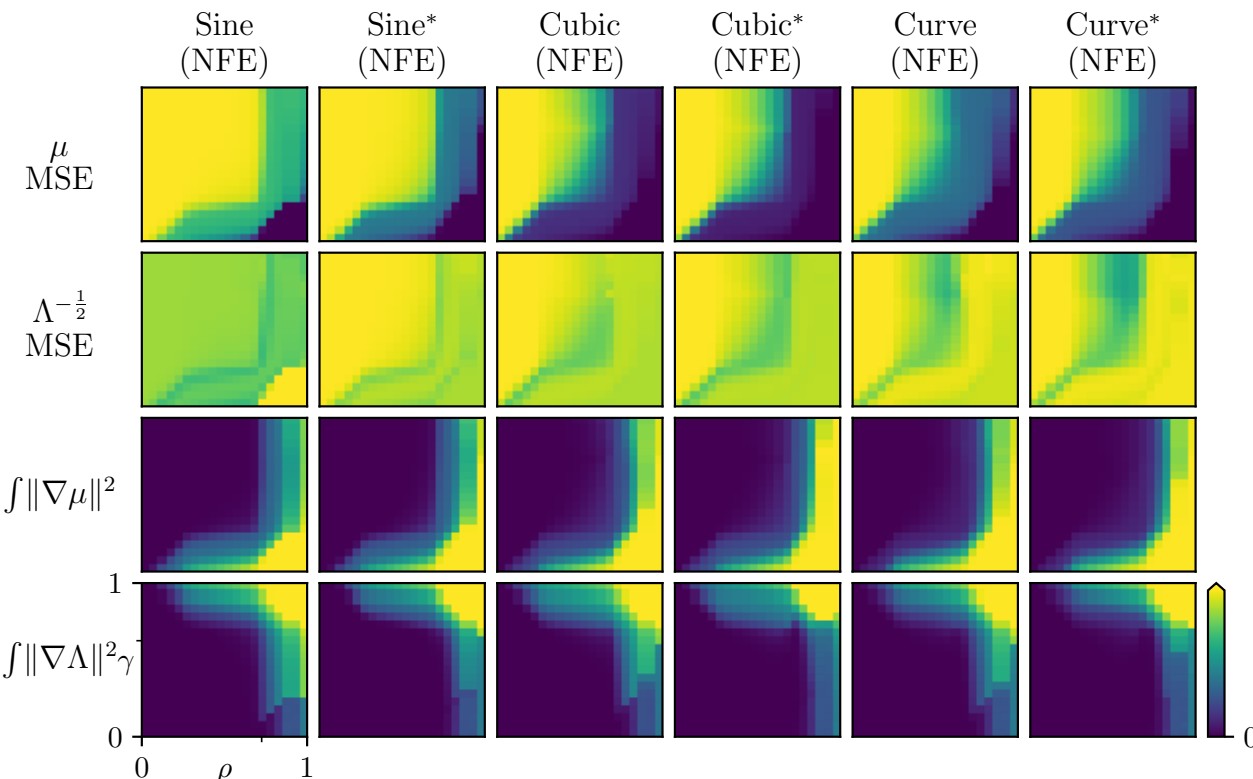

Figure 6: Same configuration as Fig. 2, except all results here pertain to minimizing the FT on six different synthetic datasets described in Table 3. Dataset names with an ∗ are the homoskedastic counterparts.

## C  ADDITIONAL RESULTS

### C.1  ALL SYNTHETIC DATASET RESULTS

Both FT and neural networks were fit to the heteroskedastic and homoskedastic synthetic datasets described in Table 3. The main results for these displayed as phase diagrams of various metrics can be seen in Fig. 6 and Fig. 7 respectively. We largely see the same trends as were exhibited by the real-world datasets seen in Fig. 2.

### C.2  EFFECT OF NEURAL NETWORK SIZE

We used the same training methods to fit models with one and two hidden layers and fit them to the *concrete* dataset. The results in the phase diagram were consistent with the other experiments, as can be seen in Fig. 8.

## D  COMPARISON TO BASELINES

We compare the performance of our diagonal $\rho + \gamma = 1$ search against two baselines, $\beta$-NLL [Seitzer et al., 2022] and an ensemble of six MLE-fit heteroskedastic regression models [Lakshminarayanan et al., 2017]. We use $\mu$ MSE, $\Lambda^{-\frac{1}{2}}$ MSE, and expected calibration error (ECE) to evaluate the models. In all cases lower values are better. Note that the method of ensembling multiple individual MLE-fit models from Lakshminarayanan et al. [2017] could be implemented on our method or $\beta$-NLL as well.

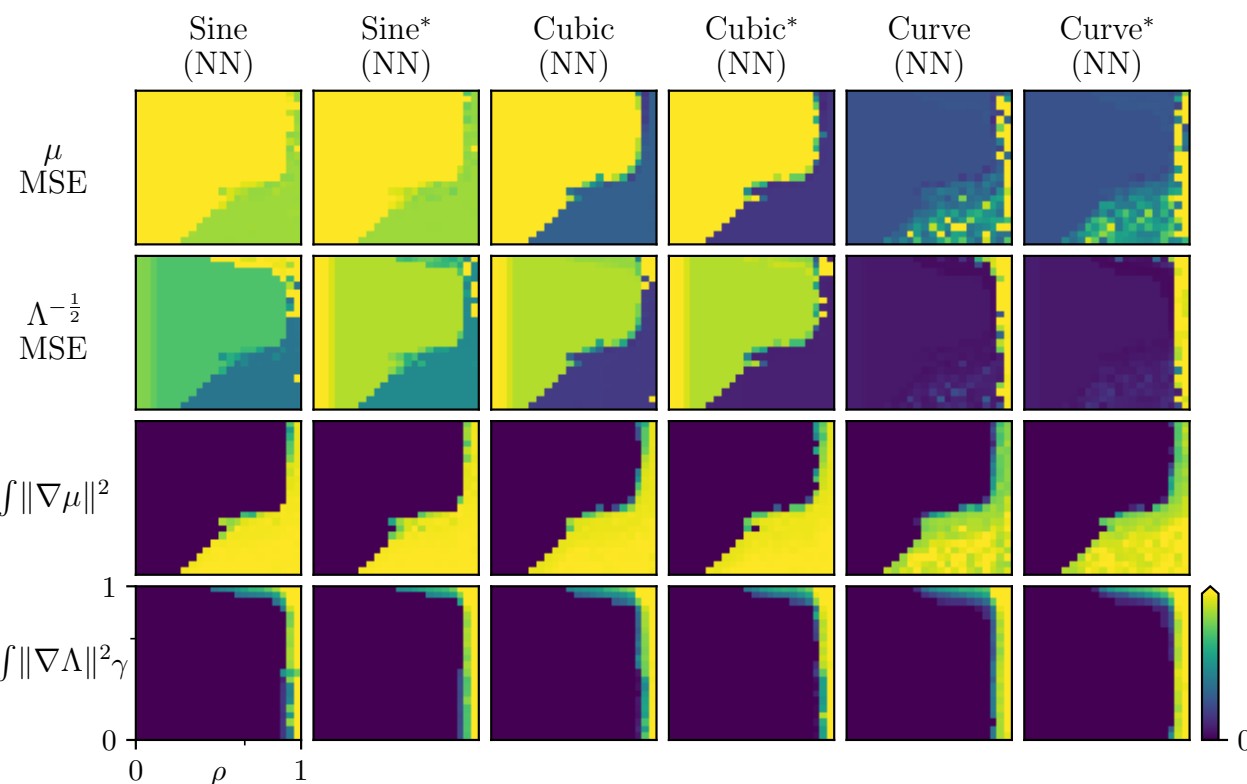

Figure 7: Same configuration as Fig. 2 and Fig. 6, except all results here pertain to training a neural network on six different synthetic datasets described in Table 3. Dataset names with an ∗ are the homoskedastic counterparts.

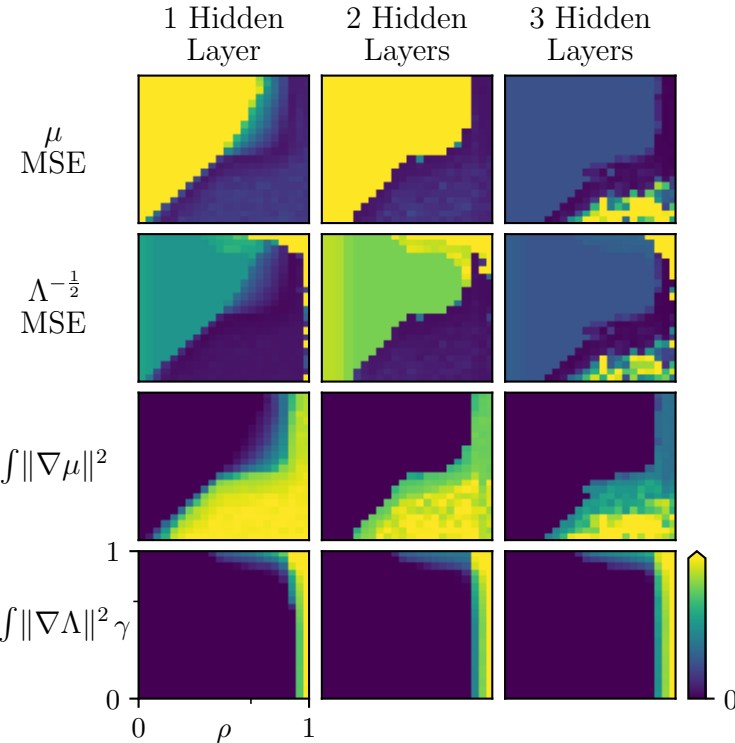

Figure 8: Same configuration as Fig. 2, however, these results pertain all to fitting a neural network of various sizes on the *concrete* dataset.

### D.1 MODEL ARCHITECTURE

All (individual) models have the same architecture: fully connected neural networks with three hidden layers of 128 nodes and leaky ReLU activations for the synthetic and UCI datasets and fully connected neural networks with three hidden layers of 256 nodes for the ClimSim data [Yu et al., 2023]. Note that both baselines model the variance while our approach models the precision (inverse-variance). In all cases we use a softplus on the final layer of the variance/precision networks to ensure the output is positive.

For the $\beta-$NLL implementation we take $\beta = 0.5$ as suggested in Seitzer et al. [2022]. The ensemble method we use fits 6 individual heteroskedastic neural networks and combines their outputs into a mixture distribution that is approximated with a normal distribution. We do not add in adversarial noise as the authors state it does not make a significant difference. We fit six $\beta-$NLL models and six MLE-ensembles.

### D.2 DIAGONAL SELECTION CRITERIA

After conducting our diagonal search we found the model that minimized $\mu$ MSE and the model that minimized $\Lambda^{-\frac{1}{2}}$ MSE on the *training* data. In some cases these models coincided. We then used the model that was on the midpoint (on a logit scale) of the $\rho + \gamma = 1$ line between these two models to compare. The results are reported in Table 5. In all cases our method is competitive with or exceeds the performance of these two baselines–particularly on real-world data. Note that our goal is to show that we are able to find models that model the mean and standard deviation of the data well, that is, lie in our proposed region $S$ of the phase diagram. We do not claim that this method will always provide optimal model within $S$.

### D.3 TRAINING DETAILS

Training for our method was conducted as described in sections B.2 and B.6 of the appendix.

For the baselines, on all of the simulated datasets we train for 600000 epochs and use the Adam optimizer with a basic

triangular cycle that scales initial amplitude by half each cycle on the learning rate. The minimum and maximum learning rates were 0.0001 and 0.01. The cycles were 50000 epochs. We clip the gradients at 1000. The same optimization scheme is performed for the UCI datasets but for $500000$ epochs for the *Housing*, *Concrete*, and *Yacht* datasets. The *Power* dataset was trained for 50000 epochs with batches of 1000.

## D.4 CLIMSIM DATASET

The ClimSim dataset [Yu et al., 2023] is a largescale climate dataset. Its input dimension is 124 and output dimension is 128. We use all 124 inputs to model a single output, *Visible direct solar flux, SOLS [W/$m^2$ ]*. We train on 10,091,520 of the approximately 100 million points for training and we use a randomly selected 7,209 points to evaluate our models.

## D.5 DEFICIENCY OF ECE

Shortcomings of ECE (in isolation) are well documented [Kuleshov et al., 2018, Levi et al., 2022, Chung and Neiswanger, 2021]. The main issue with ECE is it measures *average* calibration, while *individual* calibration is more desirable. On our diagonal search we found that often times the models that achieved the best ECE were those that were severely underfit and belonged to region $U_I$. In Table 5 we see that the MLE-ensemble is able to achieve low scores while being uncompetitive with respect to the two MSE metrics. The MLE-ensembles were unstable on several of the datasets with respect to the variance network which is consistent with Proposition 1. In particular this can be seen for the synthetic datasets the $\Lambda^{-\frac{1}{2}}$ MSE diverges to infinity.

# E   FOURIER FEATURE MAPPINGS AND GEOMETRIC COMPLEXITY

As a preprocessing step we apply Fourier feature mapping Tancik et al. [2020] before passing our data into MLPs. That is, we map inputs $x \to \gamma(x)$ where $\gamma(\cdot)$ is defined as follows

$$\gamma(x) = [\cos(2\pi b_1^T x), \dots, \cos(2\pi b_k^T x)]$$

and the $\{b_i\}_{i=1}^{k}$ are independently sampled from a $\mathcal{N}(0, 1)$ distribution. This method has been shown to allow MLPs to learn high frequency data and to reduce training time. The motivation for this additional step is to encourage the mean and precision networks to overfit and hopefully, mimic some of the behaviors of the FT more faithfully. We try this in two different settings. In the first we add in the Fourier Features layers with an L2 penalty and then again with the Dirichlet energy/geometric complexity as the regularizer similar in spirit to Hoffman et al. [2019]. Just as in the earlier experiments we penalize the mean and precision networks separately and weigh the regularizers in the same $(\rho, \gamma)$-scheme.

In the case where we penalize the geometric complexity the regularizer now matches exactly the regularizer of the field theory setting. We find even greater correspondence between results. However, there is a heavy computational burden where training takes on the order of 10 times slower in wall clock time.

We use 2-layer MLPs with width 128 and set the Fourier features mapping to be 64-dimensional, with $\sigma = 2$ when sampling the weights for both the $\mu$ and $\Lambda$ networks. We remove one layer from the MLPs (compared to the earlier experiments) to accommodate the fact that we add in the Fourier feature mapping. We train the models to 128 samples from the generated in the same way as the *Sine* dataset with 5000 epochs warmup for the mean network and 150000 epochs total and have batch size of 32. Despite fewer training epochs than the earlier neural network experiments, we still achieve overfitting behavior. We use the same set of $(\gamma, \rho)$-pairings as in the neural network experiments described in section B.5. Summary plots of $\mu$- and $\Lambda^{-1/2}$-MSE as well as Dirichlet energies can be found in Fig. 9. A subset of resulting fits can be seen in Figs. 12 and 13.

Table 5: Comparison of a deep heteroskedastic regression model with diagonal regularization search against two baselines [Seitzer et al., 2022, Lakshminarayanan et al., 2017]. For details on the selection criteria of the heteroskedastic model see Appendix D.2. We report the average and standard deviations of expected calibration error (ECE), $\mu$ MSE and $\Lambda^{-\frac{1}{2}}$ MSE on test data. Lowest mean value for each metric is bolded. In several cases the MLE ensemble failed to properly converge (yielding inf $\pm$ nan results when the standard deviation function diverges to infinity). Individually, there are many pitfalls to using MLE to train heteroskedastic regression models, and it only takes one member of the ensemble to fail to diverge to yield these results. In particular, note that these numerical issues occur most commonly for the quantities relating to the standard deviation. This highlights the instability of MLE training in this setting. Note that the method of ensembling multiple individual MLE-fit models could be performed on our method or $\beta$-NLL as well.

| Dataset
Metric | Heteroskedastic | $\beta$-NLL
Seitzer et al. [2022] | MLE Ensemble
Lakshminarayanan et al. [2017] |
|---|---|---|---|
| **Cubic** | | | |
| ECE | **0.2380** $\pm$ 0.03 | 0.2385 $\pm$ 0.02 | 0.2411 $\pm$ 0.02 |
| $\mu$ MSE | 0.2339 $\pm$ 0.01 | **0.1500** $\pm$ 0.01 | 1.1809 $\pm$ 1.88 |
| $\Lambda^{-\frac{1}{2}}$ MSE | 0.2397 $\pm$ 0.02 | **0.1397** $\pm$ 0.01 | inf $\pm$ nan |
| **Curve** | | | |
| ECE | 0.1804 $\pm$ 0.02 | **0.1754** $\pm$ 0.02 | 0.2432 $\pm$ 0.00 |
| $\mu$ MSE | **0.4318** $\pm$ 0.12 | 0.4877 $\pm$ 0.16 | 1.0067 $\pm$ 0.19 |
| $\Lambda^{-\frac{1}{2}}$ MSE | 0.4655 $\pm$ 0.09 | **0.4187** $\pm$ 0.20 | inf $\pm$ nan |
| **Sine** | | | |
| ECE | 0.2499 $\pm$ 0.00 | **0.2082** $\pm$ 0.03 | 0.2313 $\pm$ 0.05 |
| $\mu$ MSE | **0.7968** $\pm$ 0.00 | 4.4107 $\pm$ 6.90 | 0.9716 $\pm$ 0.06 |
| $\Lambda^{-\frac{1}{2}}$ MSE | **0.7968** $\pm$ 0.00 | 4.3524 $\pm$ 6.89 | inf $\pm$ nan |
| **Concrete** | | | |
| ECE | 0.2471 $\pm$ 0.01 | 0.2552 $\pm$ 0.00 | **0.0655** $\pm$ 0.01 |
| $\mu$ MSE | **0.1055** $\pm$ 0.02 | 0.5461 $\pm$ 0.30 | 2.2454 $\pm$ 1.74 |
| $\Lambda^{-\frac{1}{2}}$ MSE | **0.3028** $\pm$ 0.51 | 1.0867 $\pm$ 0.20 | $1.3 \times 10^5 \pm 1.2 \times 10^5$ |
| **Housing** | | | |
| ECE | 0.0653 $\pm$ 0.00 | **0.2631** $\pm$ 0.01 | 0.1332 $\pm$ 0.02 |
| $\mu$ MSE | **1.2236** $\pm$ 0.00 | 0.3175 $\pm$ 0.06 | 155.4494 $\pm$ 128.27 |
| $\Lambda^{-\frac{1}{2}}$ MSE | **0.7610** $\pm$ 0.00 | 0.8820 $\pm$ 0.03 | 218.8269 $\pm$ 195.38 |
| **Power** | | | |
| ECE | 0.2233 $\pm$ 0.01 | 0.2370 $\pm$ 0.00 | **0.0285** $\pm$ 0.01 |
| $\mu$ MSE | 0.0350 $\pm$ 0.01 | 0.1013 $\pm$ 0.01 | **0.0177** $\pm$ 0.00 |
| $\Lambda^{-\frac{1}{2}}$ MSE | 0.0343 $\pm$ 0.01 | 0.3081 $\pm$ 0.37 | **0.0091** $\pm$ 0.00 |
| **Yacht** | | | |
| ECE | 0.3038 $\pm$ 0.04 | 0.2882 $\pm$ 0.02 | **0.0463** $\pm$ 0.02 |
| $\mu$ MSE | **0.0077** $\pm$ 0.01 | 0.0137 $\pm$ 0.01 | 6.2670 $\pm$ 13.96 |
| $\Lambda^{-\frac{1}{2}}$ MSE | **0.0076** $\pm$ 0.01 | 1.3275 $\pm$ 0.02 | 8.0599 $\pm$ 19.18 |
| **Solar Flux** | | | |
| ECE | **0.1503** $\pm$ 0.00 | 0.3007 $\pm$ 0.00 | 0.1924 $\pm$ 0.04 |
| $\mu$ MSE | **0.2887** $\pm$ 0.00 | 0.3771 $\pm$ 0.00 | 1.0067 $\pm$ 0.19 |
| $\Lambda^{-\frac{1}{2}}$ MSE | **0.1175** $\pm$ 0.00 | 0.3217 $\pm$ 0.00 | $4.6 \times 10^9 \pm 9.9 \times 10^9$ |

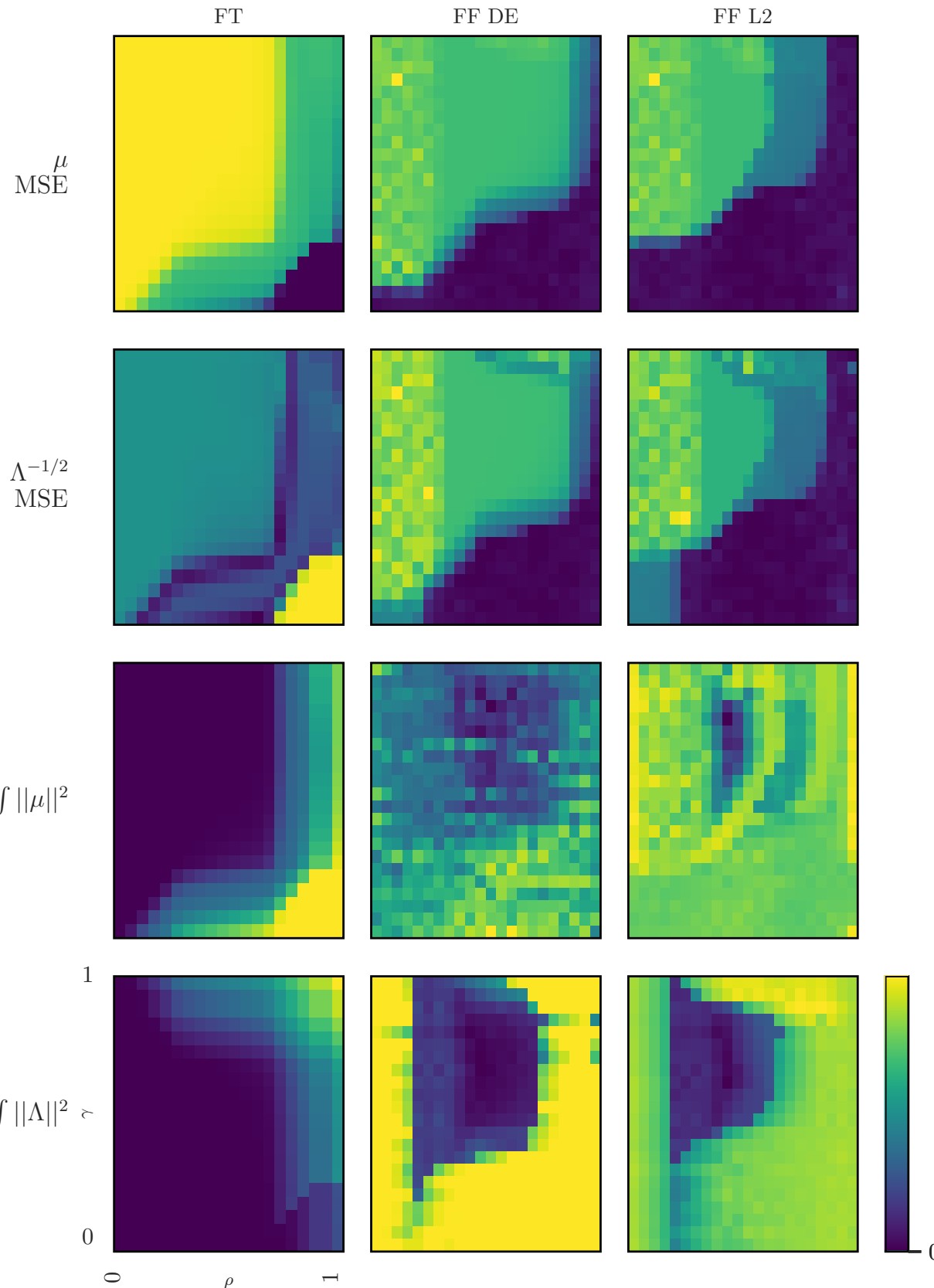

Figure 9: Phase diagrams for the field theory (left), and MLPs with Fourier feature mappings with Dirichlet energy regularization (middle) and L2 regularization (right) fit to the *Sine* dataset.

# Field Theory Fits

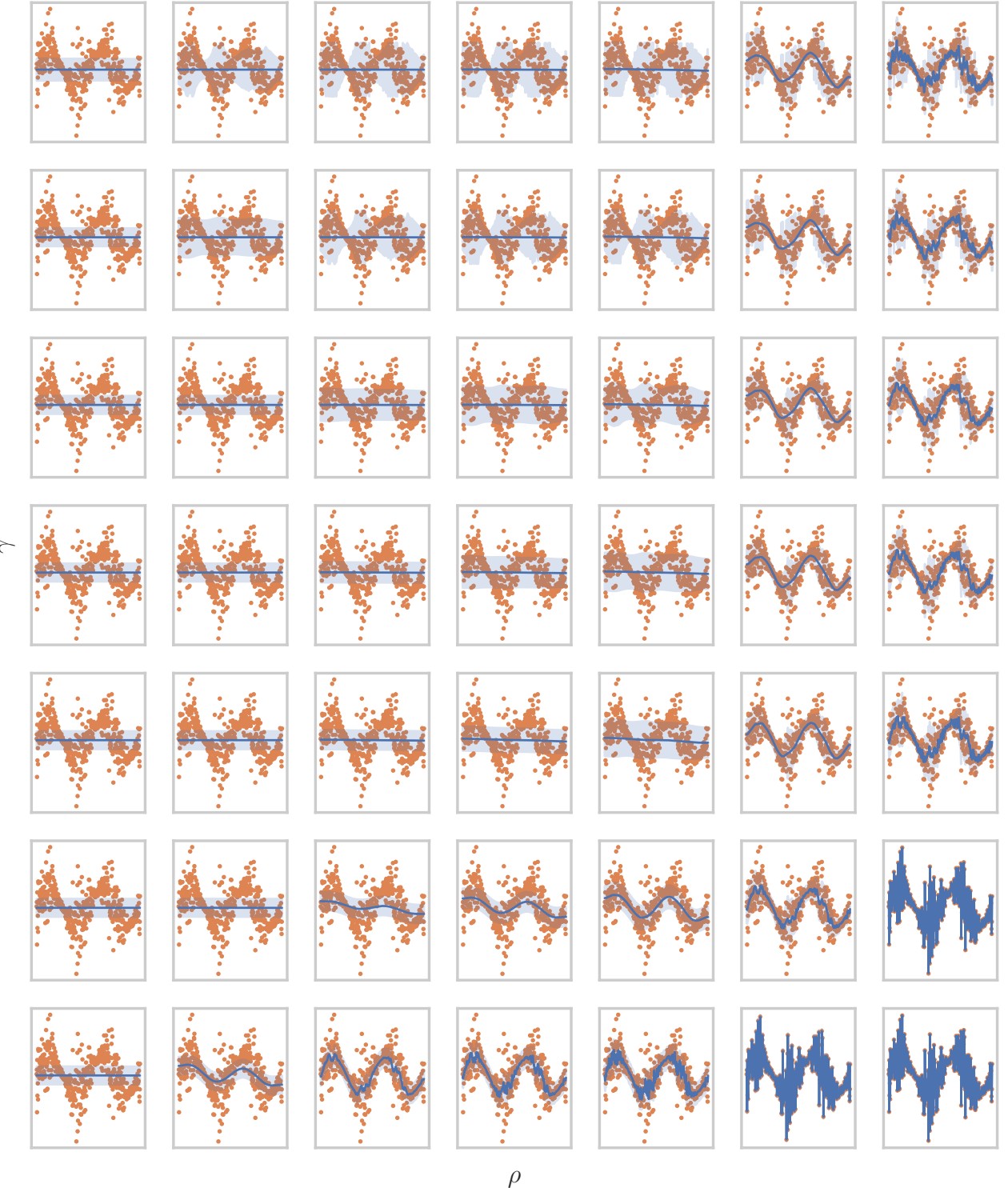

Figure 10: Subsample of fits of the field theory. Moving right to left increases $\rho$ while moving up and down to up increases $\gamma$. Training data is shown in orange, the mean function is shown in red, and $\pm 1$ SD is shaded. Note: FT was fit to 4096 datapoints, but here we display a thinned subset of the points for visual clarity.

# MLP (L2) Fits

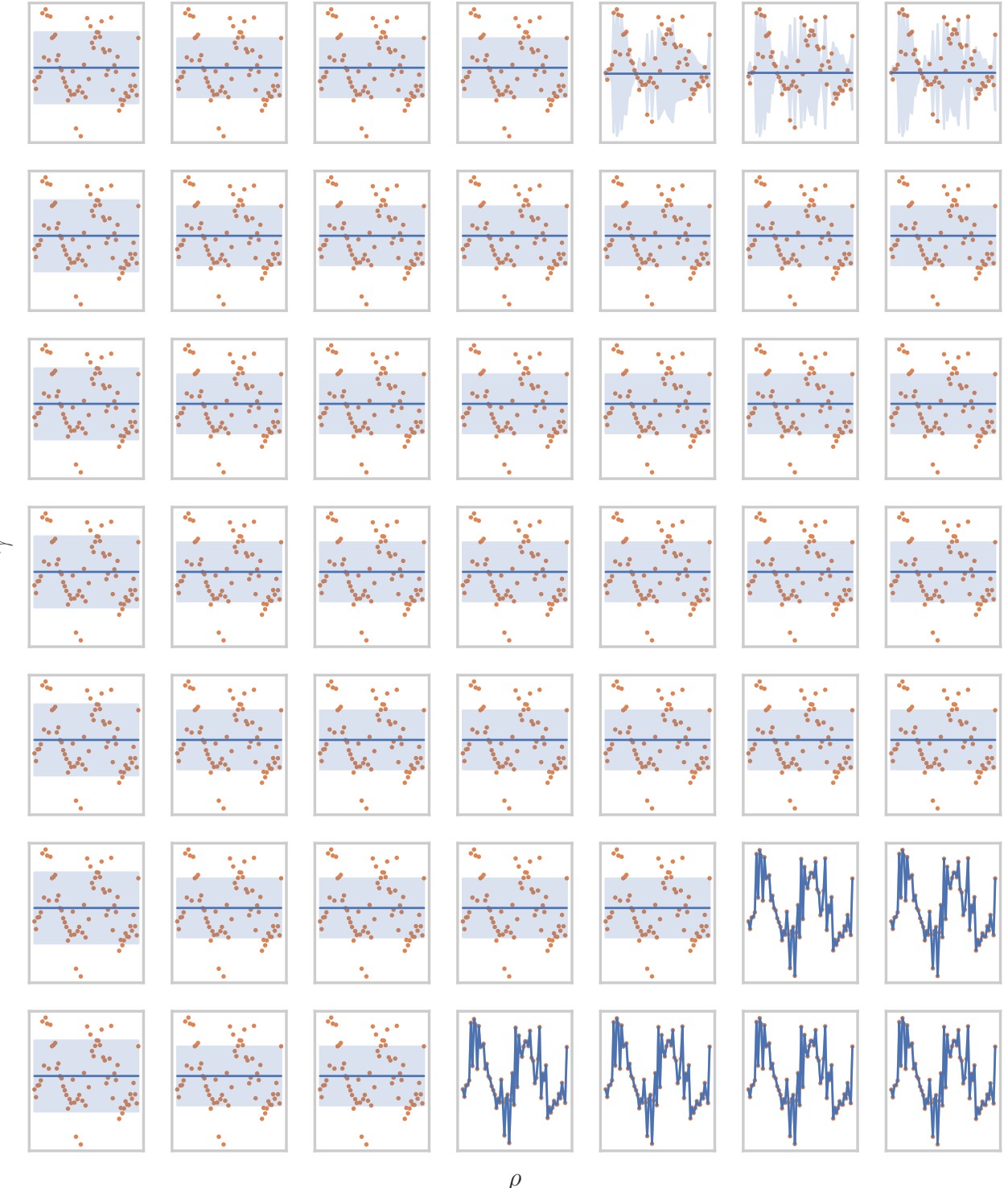

Figure 11: Subsample of fits of the neural networks on the *Sine* dataset. Training data is shown in orange, the mean function is shown in red, and ± 1 SD is shaded. Notice the abrupt phase transition from overfitting to underfitting in the mean function (in the lower right corner) and similarly in the precision function (in the upper right corner). Moving right to left increases $\rho$ while moving up and down to up increases $\gamma$.

# Fourier Feature (DE) Fits

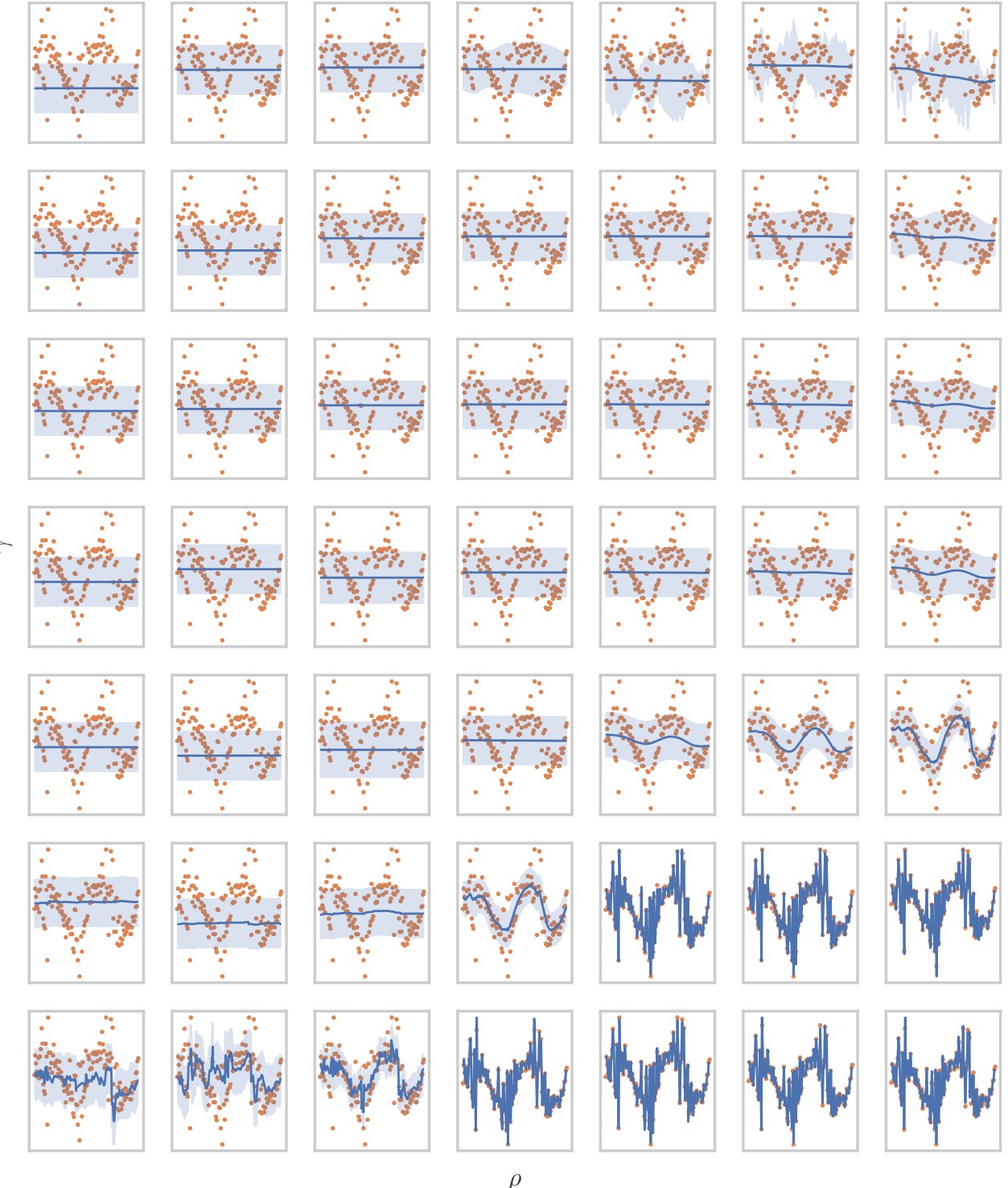

Figure 12: Subsample of fits of neural networks with Fourier Feature layers and Dirichlet energy/geometric complexity regularization. Training data is shown in orange, the mean function is shown in red, and $\pm$ 1 SD is shaded. Moving right to left increases $\rho$ while moving up and down to up increases $\gamma$.

# Fourier Feature (L2) Fits

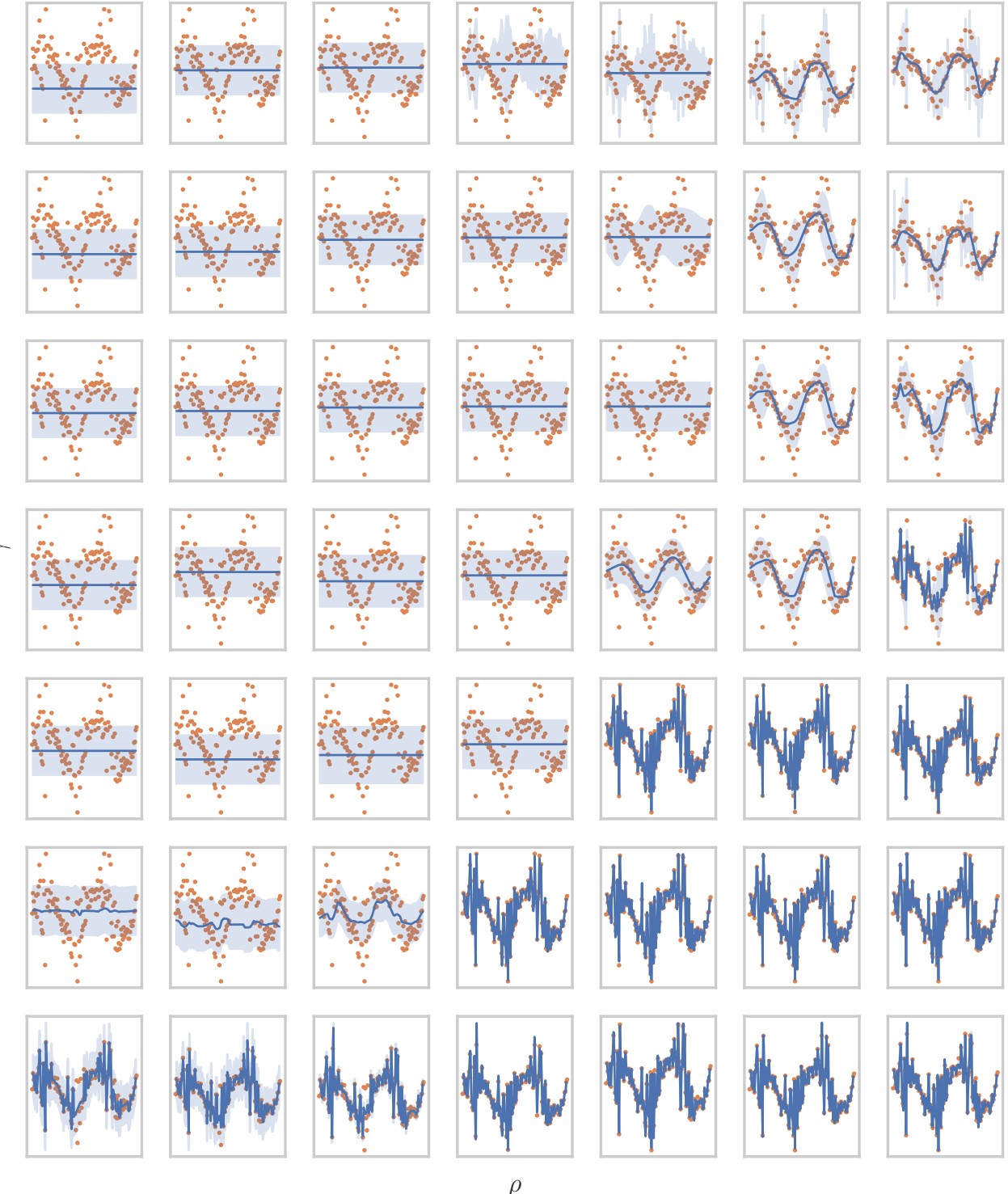

Figure 13: Subsample of fits of neural networks with Fourier Feature layers and L2 complexity regularization. Training data is shown in orange, the mean function is shown in red, and $\pm$ 1 SD is shaded. Moving right to left increases $\rho$ while moving up and down to up increases $\gamma$.