# OpenReview forum: "Understanding Pathologies of Deep Heteroskedastic Regression"
_auai.org/UAI/2024/Conference — UAI 2024 oral_

### Official Review · Reviewer_iuRR · 2024-03-15

**Q2-1 Originality-Novelty:** 2
**Q2-2 Correctness-Technical Quality:** 3
**Q2-5 Clarity Of Writing:** 3

**Q1 Summary And Contributions:**

This paper presents an empirical study of heteroskedastic regression (i.e.,
learning a mean and variance function depending on some features) using deep
neural networks. How to appropriately regularize both functions without the risk
of overfitting either component is the main contribution of this paper. In
particular, problem is tackled by field theory and the authors draw interesting
connections between both fields (especially the phase diagram in Figure 1 is
very nice). The only major criticism I have is regarding the quantitative
results: The paper does not compare against some baselines which may perform at
least as well as the proposed method; and it is not discussed/investigated why
the (only) competitor (\beta-NLL) yields MSEs _several orders of magnitude
larger_ than the authors proposed method. This may hint at insufficient tuning
of the competitor, numerical issues or bugs in the code and needs to be
addressed.

**Q2-3 Extent To Which Claims Are Supported By Evidence:**

2: Fair: the main claims are somewhat supported by evidence (but the experimental evaluation may be weak, or does not match entirely with the claims, important baselines may be missing, proofs contain important ideas but lack rigor, algorithmic details are only discussed superficially, references are imprecise, assumptions are not sufficiently motivated or explicated, etc.).

**Q2-4 Reproducibility:**

3: Good: key resources (e.g. proofs, code, data) are available and key details (e.g. proofs, experimental setup) are sufficiently well-described for competent researchers to confidently reproduce the main results.

**Q3 Main Strengths:**

* The paper is written well and accessibly. The problem is clearly motivated and
  contributions are stated adequately.

* The paper nicely bridges ideas from statistics/machine learning and physics.

* The authors provide readable code including a README on how to reproduce the
  results.

**Q4 Main Weakness:**

* Important baselines are missing.

* The \beta-NLL MSEs in Table 2 are several orders of magnitude larger than the
  proposed method which casts severe doubt on the credibility of the results.

**Q5 Detailed Comments To The Authors:**

* How is the proposed method superior to a two-step approach where one learns an
  appropriately regularized mean function first and then a variance function
  from the obtained residuals? This naive baseline should not be missing in my
  opinion.

* Another approach that the authors do not consider/discuss is to simply fit a
  mean function and use conformal prediction for assessing uncertainty using the
  obtained residuals. Although there is no explicit learning of a variance
  function, this approach comes with coverage guarantees that seem to be missing
  in this empirical study.

* I can only re-emphasize that I am wondering what is going on in Table 2. Also
  the inf \pm nan in Table 5 of the supplementary material (which are not
  discussed) make me doubt the validity of the presented empirical results.

**Q9 Complying With Reviewing Instructions:**

Yes

---

> ### Author Rebuttal · Authors · 2024-04-07
>
> Thank you for your constructive and thoughtful feedback. We appreciate the opportunity to address your concerns and provide clarity on our work.
>
> >The $\beta$-NLL MSEs in Table 2 are several orders of magnitude larger than the proposed method which casts severe doubt on the credibility of the results.
>
> This is a good observation. We re-ran the analysis and found there was an error in the previous version of the paper. We find that our method is still competitive with the proposed $\beta$-NLL baselines (pointwise up to two decimal places: 8 wins, 1 tie and 3 losses). The updated values are below:
>
> MSE
> | Dataset |  Ours | $\beta$-NLL |
> |----------|----------|----------|
> | Sine | 0.80 $\pm$ 0.00 |  **0.69** $\pm$ 0.05 |
> | Concrete |  **0.11** $\pm$ 0.02 | 0.55 $\pm$ 0.30 |
> | Housing |  1.22 $\pm$ 0.00 | **0.32** $\pm$ 0.05 |
> | Power |   **0.04** $\pm$ 0.01 | 0.09  $\pm$ 0.01 |
> | Yacht | 0.01 $\pm$ 0.01 | 0.01 $\pm$ 0.01 |
> | Solar Flux | **0.29** $\pm$ 0.00 | 0.38 $\pm$ 0.00 |
>
> SDMSE
> | Dataset |  Ours | $\beta$-NLL |
> |----------|----------|----------|
> | Sine | 0.80 $\pm$ 0.00 | **0.52** $\pm$ 0.07 |
> | Concrete | **0.30** $\pm$ 0.51 | 1.09 $\pm$ 0.20 |
> | Housing | **0.76** $\pm$ 0.00 | 0.88 $\pm$ 0.03 |
> | Power |  **0.03** $\pm$ 0.01 | 1.01 $\pm$ 0.01 |
> | Yacht | **0.01** $\pm$ 0.01 | 1.33 $\pm$ 0.02 |
> | Solar Flux | **0.12** $\pm$ 0.00 | 0.32 $\pm$ 0.00 |
>
> Edit: in the first posting of the rebuttal the MSE/SDMSE values were flipped for our method. This has since been corrected.
>
> >How is the proposed method superior to a two-step approach where one learns an appropriately regularized mean function first and then a variance function from the obtained residuals? This naive baseline should not be missing in my opinion.
>
> Thank you for the suggestion; however, this procedure will not eliminate the need of a two-dimensional grid search over regularization strengths. Note that we consider the overparameterized regime, in which an unregularized mean (or variance) function will overfit the training data. Thus, the proposed two-step approach would first require determining the mean-function’s regularization along a grid of varying strengths. For every mean regularization, different values for the variance function’s regularization need to be tested, since overfitting to the resulting residuals would not be conducive to generalized uncertainty estimates for new data points. It becomes apparent that this is really just a specific way of navigating the proposed 2D regularization grid for jointly learned mean and variance functions and, thus is already captured in the analyses presented in the paper.
>
> >Another approach that the authors do not consider/discuss is to simply fit a mean function and use conformal prediction for assessing uncertainty using the obtained residuals. Although there is no explicit learning of a variance function, this approach comes with coverage guarantees that seem to be missing in this empirical study.
>
> This is a good suggestion; we will implement and add this baseline for the final version of the paper. Since conformal prediction is a plug-and-play approach that can be applied to any trained model, we will also add a locally adaptive (heteroskedastic) conformal prediction model [1] that adds post hoc calibration to the heteroskedastic regression models we present in the paper.
>
> >...the [$\inf \pm nan$] in Table 5 of the supplementary material (which are not discussed) make me doubt the validity of the presented empirical results.
>
> The additional baseline we include in Table 5 is an ensemble of (overparameterized) MLE fit models [2]. Individually, there are many pitfalls to using MLE to train heteroskedastic regression models, and it only takes one member of the ensemble to fail to diverge to yield these results $\inf \pm nan$ results. This highlights the instability of MLE training in this setting. We will add details on this in the final version of the paper.
>
> [1] Lei, J., G’Sell, M., Rinaldo, A., Tibshirani, R. J., & Wasserman, L. (2018). Distribution-Free Predictive Inference for Regression. Journal of the American Statistical Association, 113(523), 1094–1111. https://doi.org/10.1080/01621459.2017.1307116
>
> [2] Balaji Lakshminarayanan, Alexander Pritzel, and Charles Blundell. Simple and Scalable Predictive Uncertainty Estimation using Deep Ensembles. In Neural Information Processing Systems, 2017

---

### Official Review · Reviewer_Y7La · 2024-03-20

**Q2-1 Originality-Novelty:** 3
**Q2-2 Correctness-Technical Quality:** 3
**Q2-5 Clarity Of Writing:** 3

**Q10 Ethical Concerns:**

No ethical concerns

**Q1 Summary And Contributions:**

This submission deals with the problems of fitting linear regression models where the (normally distributed) error terms do not have constat variance. Within this framework, models are fitted through deep neural networks. Furthermore, two ridge penalty terms are introduced: one for the network parameters relative to the mean and one for the network parameters relative to the variance.

Because of the overparameterization of the considered neural networks, different situations may occur depending of the values taken by the regularization coefficients. On the one side, one may obtain an excessive amount of regularization, i.e. underfitting of the data, concerning either the means or the variances or both. On the other side, a heavy overfitting can be observed. It is therefore of interest to identify the region where the fitting can be considered as adequate.

The main aim of the paper is to gain insight into the behaviour of the models resulting from different values of the regularization coefficients. Interestingly, it is shown empirically that the method is characterized by the presence of "phase transitions", defined as the sudden and discontinuous change of in certain metrics of interest. Furthermore, a field theory model is developed that seems to mimic the behaviour of heteroskedastic regression models, because it presents similar phase transitions.

The results of the paper are of theoretical interest with limited practical usefulness, although the results may be exploited to specify the grid of regularization coefficients so at to focus on the "adequate" region, with a consequent gain in computational efficiency.

**Q2-3 Extent To Which Claims Are Supported By Evidence:**

3: Good: the main claims are supported by convincing evidence (in the form of adequate experimental evaluation, proofs, (pseudo-)code, references, assumptions).

**Q2-4 Reproducibility:**

3: Good: key resources (e.g. proofs, code, data) are available and key details (e.g. proofs, experimental setup) are sufficiently well-described for competent researchers to confidently reproduce the main results.

**Q3 Main Strengths:**

I think that this is a well written paper that deals with a problem of interest. Despite the lack of practical applications of the results, I do appreciate the effort of trying to understand the behaviour of heteroskedastic regression models. In fact, I think that this type of work, providing insight into the intrinsic structure of a problem, may open the way to future new research directions.

**Q4 Main Weakness:**

I find original the idea of introducing a field theory model and interesting to observe how it can mimic the behaviour heteroskedastic regression models. On the other hand, it seems to me that the reason for its introduction is not sufficiently well motivated. Indeed, it is not clear to me what is the additional insight that this model provides with respect to the analysis of the behaviour of the neural network models. More specifically, I find not convincing the claim that the field theory method confers generality to the results, which "are not tied to a specific architecture or datset". This should be more clearly formalized and explained.

**Q5 Detailed Comments To The Authors:**

This submission deals with the problems of fitting linear regression models where the (normally distributed) error terms do not have constat variance. Within this framework, models are fitted through deep neural networks. Furthermore, two ridge penalty terms are introduced: one for the network parameters relative to the mean and one for the network parameters relative to the variance.

Because of the overparameterization of the considered neural networks, different situations may occur depending of the values taken by the regularization coefficients. On the one side, one may obtain an excessive amount of regularization, i.e. underfitting of the data, concerning either the means or the variances or both. On the other side, a heavy overfitting can be observed. It is therefore of interest to identify the region where the fitting can be considered as adequate.

The main aim of the paper is to gain insight into the behaviour of the models resulting from different values of the regularization coefficients. Interestingly, it is shown empirically that the method is characterized by the presence of "phase transitions", defined as the sudden and discontinuous change of in certain metrics of interest. Furthermore, a field theory model is developed that seems to mimic the behaviour of heteroskedastic regression models, because it presents similar phase transitions.

The results of the paper are of theoretical interest with limited practical usefulness, although the results may be exploited to specify the grid of regularization coefficients so at to focus on the "adequate" region, with a consequent gain in computational efficiency.

I think that this is a well written paper that deals with a problem of interest. Despite the lack of practical applications of the results, I do appreciate the effort of trying to understand the behaviour of heteroskedastic regression models. In fact, I think that this type of work, providing insight into the intrinsic structure of a problem, may open the way to future new research directions.

I find original the idea of introducing a field theory model and interesting to observe how it can mimic the behaviour heteroskedastic regression models. On the other hand, it seems to me that the reason for its introduction is not sufficiently well motivated. Indeed, it is not clear to me what is the additional insight that this model provides with respect to the analysis of the behaviour of the neural network models. More specifically, I find not convincing the claim that the field theory method confers generality to the results, which "are not tied to a specific architecture or datset". This should be more clearly formalized and explained.

**Q9 Complying With Reviewing Instructions:**

Yes

---

> ### Author Rebuttal · Authors · 2024-04-07
>
> Thank you for your constructive and thoughtful feedback. We appreciate the opportunity to address your concerns and provide clarity on our work.
>
> >… the reason for [the field theory] introduction is not sufficiently well motivated. Indeed, it is not clear to me what is the additional insight that this model provides…I find not convincing the claim that the field theory method confers generality to the results, which "are not tied to a specific architecture or dataset".
>
> When it comes to generality with respect to datasets: this is primarily an empirical statement, evidenced by our multiple experiments. Qualitatively, we found the same types of phase diagrams and phase transitions across all considered data sets. We empirically found that boundaries between regions of interest were similar in shape across datasets but not *quantitatively* the same, i.e., phase transitions occurred at differing levels of regularization for different data sets of different dimensionality, which is not surprising. We will update the language in the final version of the paper to clarify these differences.
>
> Generality with respect to network architectures is a deeper statement, as it connects to our field-theoretical description of the observed phenomena. The advantage of the field theory is that we are able to directly analyze the behavior of arbitrary regression functions in the presence of varying levels of regularization. By replacing sums over data points with integrals, data points by densities, L2 norms with gradient penalties, and neural network regressors with arbitrary functions, we arrived at a theoretical loss function that no longer depends on the network parameters. The mathematical basis for dealing with such functional integrals was functional analysis. We showed that this abstract description of our loss function showed qualitatively similar results compared to sufficiently overparameterized neural network architectures on one-dimensional data, where we could solve the field theory numerically. Furthermore, we showed that different neural networks showed similar phase diagrams and phase transitions on various real-world datasets from the UCI machine learning repository.
>
> >The results of the paper are of theoretical interest with limited practical usefulness, although the results may be exploited to specify the grid of regularization coefficients so at to focus on the "adequate" region, with a consequent gain in computational efficiency.
>
> We consider the main contributions of this paper to be both theoretical and practical. First we reframe a deep learning problem in the language of theoretical physics. From there we utilize non-trivial methods inspired by theoretical physics to empirically investigate the complex behavior of heteroskedastic regression models. Through analyzing empirically derived phase diagrams we were able to derive an efficient hyperparameter search strategy that yielded useful model fits in a principled fashion on UCI regression datasets and the ClimSim dataset.

---

### Official Review · Reviewer_B3yP · 2024-03-23

**Q2-1 Originality-Novelty:** 2
**Q2-2 Correctness-Technical Quality:** 2
**Q2-5 Clarity Of Writing:** 4

**Q1 Summary And Contributions:**

In this paper the authors provided a theoretical explanation of the behavior of neural network based heteroskedastic regression. The authors then compared the theoretical prediction with empirical data, and suggested an efficient way of search the hyper-parameter space.

**Q2-3 Extent To Which Claims Are Supported By Evidence:**

3: Good: the main claims are supported by convincing evidence (in the form of adequate experimental evaluation, proofs, (pseudo-)code, references, assumptions).

**Q2-4 Reproducibility:**

4: Excellent: key resources (e.g. proofs, code, data) are available and key details (e.g. proof sketches, experimental setup) are comprehensively described for competent researchers to confidently and easily reproduce the main results.

**Q3 Main Strengths:**

1. The authors provided a theoretical frame work for the understanding of the behavior of neural network based heteroskedastic regression.
2. The authors suggested an efficient way of search the hyper-parameter space.

**Q4 Main Weakness:**

The 3rd statement of proposition 1 and its proof are problematic.

**Q5 Detailed Comments To The Authors:**

The 3rd statement of proposition 1 claims that "there are no valid solutions to the FT if \rho \in (0, 1) and \gamma = 1 (should there be no mean regularization, then there needs to be at least some regularization for the precision)."
In the proof for the 3rd statement, the authors consider the case where \alpha = 0. However, given \pho < 1, by the relation between \alpha, \beta and \rho, \gamma, we have \alpha+\beta > 0, which means that if \alpha = 0, then \beta > 0. Therefore, the second part of equation (22) is invalid.
Moreover, given \rho < 1, if \gamma=1, then we have \alpha>0 and \beta=0. This combination is a valid choice under which the solution to the equation (9) can be found.
To summarize, the correct statement should be: if \rho \in (0, 1), then if there is no mean regularization, there needs to be at least some regularization for the precision. This statement is not a consequence of solving the stationary condition of the field functional, but a trivial consequence of the how \rho and \gamma are defined in term of \alpha and \beta.

**Q9 Complying With Reviewing Instructions:**

Yes

---

> ### Author Rebuttal · Authors · 2024-04-07
>
> Thank you for your constructive and thoughtful feedback. We appreciate your attention to detail and have reworked the proof for part (iii) of Proposition 1 and updated the statement. We believe some of the confusion came from moving between the $(\alpha, \beta)$- and $(\gamma, \rho)$-parameterizations of the regularizers. In doing so we were able to arrive at a stronger statement. The new statement and proof are as follows:
>
> Claim: (iii) In order for there to exist a solution to the FT there must be regularization on the mean function.
>
> Proof: In the $(\alpha, \beta)$-regularization, it is equivalent to say $\alpha>0$ is a necessary condition for there to exist a solution to the FT. Recall that we seek to minimize
>
> $$\mathcal{L}_{\alpha, \beta}(\hat{\mu},\hat{\Lambda}) = \int p(x)(-\log \hat p(y | x)  + \alpha ||\nabla\hat{\mu}(x)||_2^2  + \beta ||\nabla\hat{\Lambda}(x)||_2^2) dx$$
>
> where $\hat p(y | x) = \mathcal{N}(y | \hat\mu(x), \hat\Lambda(x))$.
> Suppose $\alpha = 0$. Then the functional simplifies to
>
> $$\min_{\hat \mu, \hat \Lambda} \mathcal{L}(\hat{\mu},\hat{\Lambda}) = \min_{\hat \mu, \hat \Lambda} \int_{\mathcal{X}}  p(x)(-\log \hat p(y | x) + \beta ||\nabla\hat{\Lambda}(x)||_2^2) dx$$
>
> $$ \phantom{\min_{\hat \mu, \hat \Lambda} \mathcal{L}(\hat{\mu},\hat{\Lambda})} \leq \min_{\hat \mu, \hat {\underline \Lambda}} \int_{\mathcal{X}} p(x) (\frac{1}{2}(\hat{\underline \Lambda}(x)(\hat \mu(x) - y(x))^2 - \log \hat{\underline{\Lambda(x)}}) + \beta ||\nabla\hat{\underline\Lambda}(x)||_2^2) dx$$
>
> $$ \phantom{\min_{\hat \mu, \hat \Lambda} \mathcal{L}(\hat{\mu},\hat{\Lambda})} = \min_{\hat \mu, \hat {\underline \Lambda}} \int_{\mathcal{X}} p(x) (\frac{1}{2}(\hat{\underline \Lambda}(x)(\hat \mu(x) - y(x))^2 - \log \hat{\underline{\Lambda(x)}}) dx$$
>
> where $\hat{\underline\Lambda}$ is a constant function.
> This provides an upper bound on the integral as we are looking at a restricted class of possible precision functions.
> Since the precision function is constant the gradient penalty, $||\nabla\hat{\Lambda}(x)||_2^2$, is zero.
> There is no penalty on $\hat \mu$ so it can perfectly pass through every data point and the contribution of $\hat{\underline \Lambda}(x)(\hat \mu(x) - y(x))^2$ is zero while $- \log \hat{\underline{\Lambda}}(x)$ can become arbitrarily negative.
> Thus there is no solution if $\alpha=0$.

---

### Meta-Review · Area_Chair_T2rA · 2024-04-15

Overall a good paper with only positive evaluations. The topic is of general interest in the field and the paper points out some surprising aspects of a model family that is popular (deep overparameterized models for regression problems), and would benefit for broad exposure for the audience of the conference.